# LATE-TO-EARLY TRAINING: LET LLMS LEARN EARLIER, SO FASTER AND BETTER

**Ji Zhao**[1,2*]  **Yufei Gu**[1]  **Shitong Shao**[1]  **Xun Zhou**[2]  **Liang Xiang**[2]  **Zeke Xie**[1,†]

The Hong Kong University of Science and Technology (Guangzhou)[1]    ByteDance Seed[2]

## ABSTRACT

As Large Language Models (LLMs) achieve remarkable empirical success through scaling model and data size, pretraining has become increasingly critical yet computationally prohibitive, hindering rapid development. Despite the availability of numerous pretrained LLMs developed at significant computational expense, a fundamental real-world question remains underexplored: *Can we leverage existing small pretrained models to accelerate the training of larger models?* In this paper, we propose a Late-to-Early Training (LET) paradigm that enables LLMs to explicitly learn later knowledge in earlier steps and earlier layers. The core idea is to guide the early layers of an LLM during early training using representations from the late layers of a pretrained (i.e. late training phase) model. We identify two key mechanisms that drive LET's effectiveness: late-to-early-step learning and late-to-early-layer learning. These mechanisms significantly accelerate training convergence while robustly enhancing both language modeling capabilities and downstream task performance, enabling faster training with superior performance. Extensive experiments on 1.4B and 7B parameter models demonstrate LET's efficiency and effectiveness. Notably, when training a 1.4B LLM on the Pile dataset, our method achieves up to 1.6× speedup with nearly 5% improvement in downstream task accuracy compared to standard training, even when using a pretrained model with 10× fewer parameters than the target model.

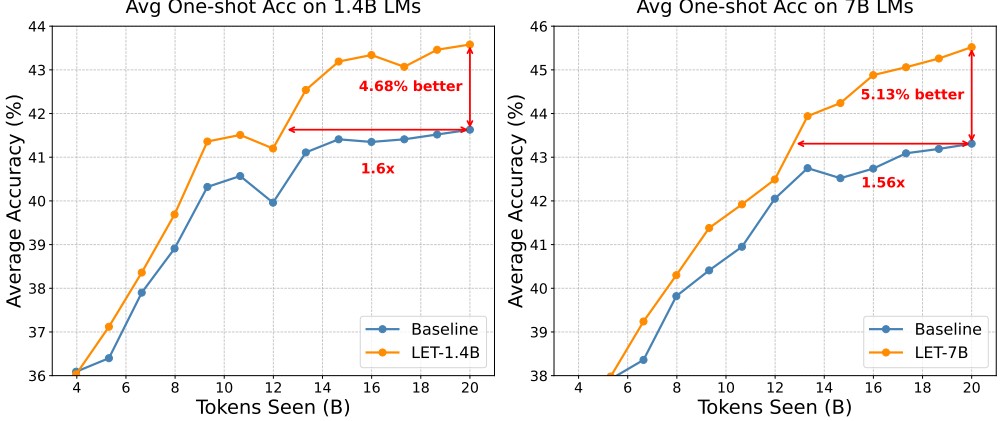

Figure 1: Comparison of Average Downstream Task Performance: LET vs. Baseline (Standard Training) on 1.4B and 7B Models. LET models are trained under our proposed LET paradigm, whereas the baseline models utilize standard causal language modeling. Remarkably, LET delivers significant performance gains, even when aligned with a model 10× smaller than the target model.

*Work done at ByteDance Seed.    †Corresponding author: zekexie@hkust-gz.edu.cn

# 1 INTRODUCTION

Large language models (LLMs) have demonstrated remarkable performance across diverse natural language tasks (Brown et al., 2020; Achiam et al., 2023; Team et al., 2023), marking a significant milestone toward artificial general intelligence (AGI) (Goertzel, 2014; Bubeck et al., 2023). Pretraining plays a key role in shaping these models' capabilities (Devlin et al., 2019; Radford et al., 2019), serving as the foundation for their downstream performance. However, training such models remains extremely resource-intensive (Kaplan et al., 2020; Rae et al., 2021; Hoffmann et al., 2022). For example, training an LLM with 12B parameters can require about 72,000 GPU hours using NVIDIA A100 GPUs (Biderman et al., 2023). which calls for more efficient training paradigms.

Meanwhile, fueled by the open-source culture within the AI community, we are witnessing a flourishing era rich with an array of publicly available models of varying sizes (Grattafiori et al., 2024; Yang et al., 2025; Guo et al., 2025). Building on open-source implementations, many impactful works have emerged by fine-tuning existing models such as Taori et al. (2023) in the text domain and Liu et al. (2023a) in the multimodal domain. This approach effectively leverages the substantial computational resources already invested in the development of these models.

Traditional knowledge distillation (KD) typically trains a smaller student model under the guidance of a more capable teacher model (Hinton et al., 2015; Romero et al., 2014). Nevertheless, in the context of LLMs, employing a substantially larger teacher inevitably incurs considerable memory and computational overhead. Furthermore, in conventional KD, student models tend to lag behind their teachers in performance, which limits their utility as a foundation for scaling LLM capabilities.

Recently, Rawat et al. (2024) claimed that smaller models can bootstrap the pretraining of larger LLMs. However, their method has notable limitations. The size gap between the teacher and the student is modest (only $1.87\times$), which limits practical applicability because the teacher remains relatively large and incurs substantial memory overhead. Moreover, the approach relies on heavy data preprocessing and underutilizes existing open-source models that were trained at considerable computational cost. Another line of research using smaller models to accelerate larger model training focuses on model growth strategies, leveraging open-source LLMs to accelerate the training of larger models (Du et al., 2024; Samragh et al., 2024; Wang et al., 2023b). While these approaches can reduce training time, they typically require deliberate architectural modifications, such as carefully calibrated increases in network depth and width, which add complexity and constrain the range of feasible architectures. Consequently, their practical utility is also limited.

This raises natural and practical questions: Given the abundance of small, pretrained open-source models, can they be generally leveraged during the pretraining of larger LLMs to guide and accelerate the learning process? Furthermore, could the larger target model learn to adaptively process and refine these representations as it progressively develops greater capabilities?

To address these questions, we propose **Late-to-Early Training (LET)**, a novel and general paradigm for enhancing LLM pretraining using the representations of small, pretrained models that were developed at considerable computational expense by the community. LET is architecture-agnostic as it relies solely on representations of LLMs rather than specific architectural constraints. Furthermore, LET is designed to remain effective despite the performance limitations of the smaller models in the later stages of LET training: As training progresses, the larger target model rapidly improves in overall capability and may eventually surpass the smaller model in overall performance, thereby reducing the effectiveness of the representations alignment. To address this, LET aligns the representations of the smaller trained model with the early layers of the target model, allowing the subsequent layers to naturally adapt to and refine these representations through learning dynamics (see Section 3.3). Extensive experiments with 1.4B, 3B, and 7B parameter models demonstrate the effectiveness and efficiency of the LET paradigm, with comparative results for 1B and 7B models shown in Figure 1. The primary contributions can be summarized as follows.

**First**, we are the first to tackle a novel, valuable, yet overlooked problem: Given the abundance of small, pretrained models developed at significant computational expense by the community, can they be leveraged to generally accelerate the pretraining process of much larger LLMs (e.g., $10\times$), regardless of LLM architectures?

**Second**, we propose the novel LET paradigm. At its core, it enables the early layers of the target model during early training steps to learn from the late layers of a smaller pretrained LLM (i.e., from

its late training phase). We identify two key mechanisms, Late-to-Early Step Learning and Late-to-Early Layer Learning, which are robust to the limitations of smaller models' representations.

**Third**, extensive experiments demonstrate that **LET** achieves both faster training and superior downstream performance. Notably, for training a 1.4B model on the Pile dataset (as shown in Figure 1), our method delivers up to a $1.6\times$ faster improvement in downstream performance compared with standard training, even when relying on a small model with up to $10\times$ fewer parameters than the target model, which significantly exceeds the typical scope of conventional knowledge distillation.

## 2 METHODOLOGY

In this section, we formally propose the LET paradigm for faster and better LLM training.

**Notation**    We introduce the notation used in the standard pretraining paradigm for LLMs. Let $\mathcal{M}$ denote an LLM with parameters $\theta$, and let $\mathbf{x} = [x_1, x_2, \ldots, x_T]$ represent an input token sequence of length $T$. The objective of pretraining is to maximize the likelihood of the sequence under $\mathcal{M}$, typically by training the model to predict each token given its preceding tokens. Formally, at each position $t$ ($1 \leq t \leq T$), the model $\mathcal{M}$ produces a conditional distribution $P_{\mathcal{M}}(x_t \mid x_{<t})$, where $x_{<t} = [x_1, \ldots, x_{t-1}]$ denotes the prefix context. We denote by $\mathcal{F}_{\mathcal{M}}^{(l)}$ the transformation implemented by the $l$-th layer of the model $\mathcal{M}$ (and analogously $\mathcal{F}_{\mathcal{T}}^{(l)}$ for a model $\mathcal{T}$), where $1 \leq l \leq L$ for a model with $L$ layers. Thus, a forward pass through the first $k$ layers of $\mathcal{M}$ is written as: $h_{\mathcal{M}}^{(k)} = \mathcal{F}_{\mathcal{M}}^{(k)} \circ \mathcal{F}_{\mathcal{M}}^{(k-1)} \circ \cdots \circ \mathcal{F}_{\mathcal{M}}^{(1)}(e_{1:T})$, yielding the hidden states after the $k$-th layer.

We propose the **LET** paradigm, which incorporates an additional alignment mechanism to guide the early training of a larger model $\mathcal{M}$ with the help of a smaller pretrained model $\mathcal{T}$. LET comprises two components: *Late-to-early-layer learning*: encouraging the early-layer representations of $\mathcal{M}$ to align with the late-layer representations of $\mathcal{T}$; *Late-to-early-step learning*: employing a pretrained model $\mathcal{T}$ (representing a later training stage) during the initial training steps, and gradually phasing it out as training progresses. We summarize the procedure of the LET paradigm in Algorithm 1.

The traditional training objective for $\mathcal{M}$ is to minimize the cross-entropy loss, i.e., the negative log-likelihood (NLL) of the target tokens over the training dataset. For a given sequence $\mathbf{x}$, this loss is formulated as:

$$\mathcal{L}_{\text{NLL}} = -\sum_{t=1}^{T} \log P_{\mathcal{M}}\big(x_t \mid x_{<t}\big). \tag{1}$$

This loss measures how well the model $\mathcal{M}$ predicts the token $x_t$ at each step $t$; a lower value indicates more accurate predictions.

In contrast to standard pretraining, knowledge distillation (KD) is a classical approach in which a smaller (or less capable) student model is trained to match the output probability distributions of a larger teacher model. As discussed in detail in Section C, in the context of language modeling, given a pretrained teacher model $\mathcal{T}$ producing soft predictions $P_{\mathcal{T}}(x_t \mid x_{<t})$, the KD loss is defined as

$$\mathcal{L}_{\text{KD}} = -\sum_{t=1}^{T} \sum_{v \in \mathcal{V}} P_{\mathcal{T}}(v \mid x_{<t}) \log P_{\mathcal{M}}(v \mid x_{<t}), \tag{2}$$

where $\mathcal{V}$ denotes the vocabulary and $v$ indexes individual tokens. This objective minimizes the cross-entropy between the teacher's and the student's predicted distributions.

Consider an input token sequence $\mathbf{x} = [x_1, x_2, \ldots, x_T]$ of length $T$, where each token $x_t$ belongs to the vocabulary $\mathcal{V}$. Let

$$e_{1:T} = [e_1, e_2, \ldots, e_T], \quad e_t \in \mathbb{R}^d$$

denote the corresponding token embeddings, with $d$ being the embedding dimension. These embeddings are processed by two models: a *target model* $\mathcal{M}$ and a *small pretrained model* $\mathcal{T}$. Let $L_{\mathcal{M}}$ and $L_{\mathcal{T}}$ denote the total number of Transformer layers in $\mathcal{M}$ and $\mathcal{T}$, respectively. The hidden states

---

**Algorithm 1** Late-to-Early Training

---

1: **Input**: Training dataset $\mathcal{D}$; target model $\mathcal{M}$; small pretrained model $\mathcal{T}$; initial projection weight $\lambda_0$; projection stop step $S_{\text{stop}}$
2: **Output**: Pretrained target model $\mathcal{M}$

3: **for** each minibatch $\mathbf{x} \sim \mathcal{D}$ **do**
4:     Forward $\mathbf{x}$ through $\mathcal{M}$ and $\mathcal{T}$ to obtain hidden representations
5:     Compute standard loss $\mathcal{L}_{\text{NLL}}$ from $\mathcal{M}$
6:     Retrieve $h_{\mathcal{M}}^{(k)}$ from layer $k$ of $\mathcal{M}$ and $h_{\mathcal{T}}^{(L_{\mathcal{T}})}$ from the final layer of $\mathcal{T}$
7:     **if** $d_{\mathcal{T}} \neq d_{\mathcal{M}}$ **then**
8:         Project $h_{\mathcal{M}}^{(k)}$ to match $h_{\mathcal{T}}^{(L_{\mathcal{T}})}$
9:     **else**
10:         Use $h_{\mathcal{M}}^{(k)}$ directly to align with $h_{\mathcal{T}}^{(L_{\mathcal{T}})}$
11:     **end if**
12:     Normalize hidden states and compute projection loss $\mathcal{L}_{\text{proj}} = -\tilde{h}_{\mathcal{M}}^{(k)\top}\tilde{h}_{\mathcal{T}}^{(L_{\mathcal{T}})}$
13:     Update projection weight $\lambda = \lambda_0 \cdot \max\left(0, \frac{S_{\text{stop}} - s}{S_{\text{stop}}}\right)$, where $s$ is the current training step
14:     Compute total loss $\mathcal{L}_{\text{total}} = \mathcal{L}_{\text{NLL}} + \lambda\mathcal{L}_{\text{proj}}$
15:     Backpropagate and update parameters of $\mathcal{M}$
16: **end for**

---

after the final layer of $\mathcal{T}$ and after the $k$-th layer of $\mathcal{M}$ are:

$$
\begin{aligned}
h_{\mathcal{T}}^{(L_{\mathcal{T}})} &= \mathcal{F}_{\mathcal{T}}^{(L_{\mathcal{T}})} \circ \mathcal{F}_{\mathcal{T}}^{(L_{\mathcal{T}}-1)} \circ \cdots \circ \mathcal{F}_{\mathcal{T}}^{(1)}(e_{1:T}), \\
h_{\mathcal{M}}^{(k)} &= \mathcal{F}_{\mathcal{M}}^{(k)} \circ \mathcal{F}_{\mathcal{M}}^{(k-1)} \circ \cdots \circ \mathcal{F}_{\mathcal{M}}^{(1)}(e_{1:T}),
\end{aligned}
\tag{3}
$$

where $\mathcal{F}^{(l)}$ denotes the transformation implemented by the $l$-th Transformer layer, and $1 \leq k \leq L_{\mathcal{M}}$. For clarity, we illustrate the case of a single token: $h_{\mathcal{T}}^{(L_{\mathcal{T}})} \in \mathbb{R}^{d_{\mathcal{T}}}$ and $h_{\mathcal{M}}^{(k)} \in \mathbb{R}^{d_{\mathcal{M}}}$ represent the hidden states from $\mathcal{T}$ and $\mathcal{M}$, respectively. When $d_{\mathcal{T}} \neq d_{\mathcal{M}}$, a projection is applied before alignment (details in Appendix G). The representations are then normalized, and the projection loss is defined as the negative cosine similarity between them:

$$
\mathcal{L}_{\text{proj}} = -\tilde{h}_{\mathcal{M}}^{(k)\top}\tilde{h}_{\mathcal{T}}^{(L_{\mathcal{T}})} = -\left(\frac{h_{\mathcal{M}}^{(k)}}{\|h_{\mathcal{M}}^{(k)}\|}\right)^{\top}\left(\frac{h_{\mathcal{T}}^{(L_{\mathcal{T}})}}{\|h_{\mathcal{T}}^{(L_{\mathcal{T}})}\|}\right).
\tag{4}
$$

To control the influence of this auxiliary alignment term during training, we introduce a weight $\lambda$ that decays linearly to zero:

$$
\mathcal{L}_{\text{total}} = \mathcal{L}_{\text{NLL}} + \lambda\,\mathcal{L}_{\text{proj}} = \mathcal{L}_{\text{NLL}} + \lambda_0 \cdot \max\left(0, \frac{S_{\text{stop}} - s}{S_{\text{stop}}}\right)\mathcal{L}_{\text{proj}}.
\tag{5}
$$

where $\lambda_0$ is the initial projection loss weight, $s$ is the current training step, and $S_{\text{stop}}$ is the step at which $\lambda$ decays to zero. This formulation implements the late-to-early-layer learning mechanism in the LET paradigm.

In the early stage of training, $\lambda$ is relatively large, allowing the model to leverage additional representational guidance from the model $\mathcal{T}$. As training progresses, $\lambda$ gradually decays according to a predefined schedule, ensuring that the model focuses on optimizing the primary objective $\mathcal{L}_{\text{NLL}}$. Overall, **LET** incorporates both late-to-early-layer learning and late-to-early-step learning into LLM pretraining, thereby promoting faster convergence and better generalization, as demonstrated by the experimental results in Section 3.

## 3 EMPIRICAL ANALYSIS

In the following paragraph, we empirically studied the proposed **LET** with various settings.

Table 1: Results on downstream evaluation datasets used in Groeneveld et al. (2024). We report accuracy scores for each task and the average across all datasets, with the best score per model size **boldfaced**. Notably, in the 1.4B scale setting, LET not only achieves higher final accuracy, but also exceeds the baseline's average performance while requiring less than 67% of the training steps even with $10\times$ smaller model $\mathcal{T}$. Here, LET (67%) denote models trained with 67% of the total training steps, using our proposed LET.

|  | ARC-c | ARC-e | HS | LAMB | OBQA | PIQA | SciQ | Wino. | BoolQ | Avg. |
|---|---|---|---|---|---|---|---|---|---|---|
| **Model Size = 1.4B** | | | | | | | | | | |
| Baseline | 17.8 | 44.2 | **28.6** | 24.1 | 26.0 | 61.5 | 73.3 | 51.4 | 47.9 | 41.6 |
| RKD | 18.0 | 42.9 | 27.7 | 24.8 | 26.3 | 62.4 | 63.7 | 52.3 | 54.8 | 41.4 |
| SALT[1] | 18.1 | 45.5 | 28.5 | 24.5 | 26.3 | 64.0 | 73.6 | 52.7 | 52.9 | 42.9 |
| LET(67%) | 17.8 | **45.7** | 28.1 | 23.8 | 26.6 | **64.6** | 72.2 | 52.6 | 51.1 | 42.5 |
| **LET** | **18.3** | 45.3 | 28.4 | **24.9** | **26.8** | 64.4 | **74.0** | **53.0** | **57.3** | **43.6** |
| **Model Size = 7B** | | | | | | | | | | |
| Baseline | 19.4 | 45.6 | 29.3 | 25.5 | 28.0 | 63.3 | 74.5 | 52.7 | 51.4 | 43.3 |
| RKD | 19.8 | 41.6 | 28.8 | 26.5 | 30.8 | 61.3 | 63.9 | 51.4 | 55.6 | 42.2 |
| SALT[1] | 19.1 | 46.8 | **30.5** | 27.4 | 30.6 | 62.1 | 76.0 | 52.9 | 56.9 | 44.7 |
| LET(67%) | 18.4 | 45.9 | 29.5 | 27.0 | 29.7 | 61.8 | 74.1 | 51.4 | **57.3** | 43.9 |
| **LET** | **20.0** | **47.4** | 29.8 | **28.6** | **31.4** | **65.3** | **76.7** | **54.4** | 55.9 | **45.5** |

## 3.1 EXPERIMENTAL SETUP

**Model Architecture**  Our models are based on the LLaMA architecture. We adopt RMSNorm and SwiGLU activations (Zhang & Sennrich, 2019; Shazeer, 2020; Touvron et al., 2023), and all models are trained using BF16 precision. In our experiments, the models $\mathcal{T}$ are drawn from the OPT family (Zhang et al., 2022), the Pythia family (Biderman et al., 2023), and the SmolLM family (Allal et al., 2025). Detailed model hyperparameters are summarized in Section F.

**Data**  We pretrain our models on The Pile dataset, a large-scale, diverse, and high-quality English text corpus designed for training large language models (Gao et al., 2020). It contains approximately 825 GB of text from 22 different sources, and our experiments use approximately 20 billion tokens.

**Pretraining Setting**  We follow the hyperparameter configuration for The Pile dataset from Rawat et al. (2024). Specifically, we use a total batch size of 2048 and an input sequence length of 1280. All experiments are conducted on 32 NVIDIA A100 80GB GPUs. We employ the AdamW optimizer (Loshchilov & Hutter, 2017) and a cosine learning rate schedule, with a linear warmup during the first 10% of training steps and a decay to 10% of the peak learning rate thereafter. Following the Groeneveld et al. (2024) setup, the peak learning rate is set to $4 \times 10^{-4}$ for 1B scale models and $3 \times 10^{-4}$ for 7B scale models. For more details, please refer to Appendix B.

**Evaluation**  We evaluate one-shot performance on the nine downstream test datasets used in Groeneveld et al. (2024); Gu et al. (2024). For more details on these tasks, please refer to Appendix K. Additionally, we report the language modeling loss on a test set from The Pile.

**Baseline Setting**  We compare the proposed LET paradigm with both the traditional causal language modeling approach (referred to as the Baseline in this paper), SALT (Rawat et al., 2024) and Reverse Knowledge Distillation (RKD). For more details, please refer to Appendix B.

---

[1]Since data selection is orthogonal to our method, we adopt SALT without data selection for a more controlled comparison.

## 3.2 MAIN RESULTS

**LET Improves Downstream Task Performance**   We empirically evaluate the effectiveness of our proposed LET paradigm by pretraining language models with 1.4B and 7B parameters and assessing their downstream task performance on the evaluation datasets used in Groeneveld et al. (2024). Additional experimental results and discussions are provided in Appendix D. As shown in Table 1, LET consistently outperforms the baseline on the majority of tasks across both scales, yielding higher average accuracy with a notable margin. These findings demonstrate that integrating both late-to-early-layer learning and late-to-early-step learning into LLM pretraining can effectively enhance generalization across downstream applications. Furthermore, in the 1.4B parameter configuration, LET employs a small pretrained model $\mathcal{T}$ that is an order of magnitude smaller ($10\times$) than the target model $\mathcal{M}$, yet it still achieves substantial performance gains over the baseline.

Compared to LET, RKD shows clear limitations when the model $\mathcal{T}$ is significantly smaller than the target model; specifically, it underperforms the baseline in both the 1.4B and 7B settings. RKD's results also exhibit certain patterns. For instance, it performs relatively well on tasks such as ARC-c and LAMB, indicating stronger reasoning abilities. However, on tasks like SciQ, which involve multiple-choice question answering in the science domain, RKD's performance is markedly lower than that of other methods. This suggests that while the distillation process may strengthen certain specific capabilities, it can also considerably hinder the model's overall learning effectiveness.

From Table 1, it is evident that RKD struggles when the teacher model is significantly smaller than the student model; specifically, it underperforms the baseline in both the 1.4B and 7B model settings. The results of RKD also exhibit certain patterns—for example, it performs relatively well on tasks such as ARC-c and LAMB, demonstrating strong reasoning ability. However, on tasks like SciQ, which focuses on multiple-choice question answering in the science domain, RKD's performance is substantially lower than that of other methods. This suggests that while the distillation process may reinforce certain capabilities in the student model, it can also significantly impede the model's overall learning capacity.

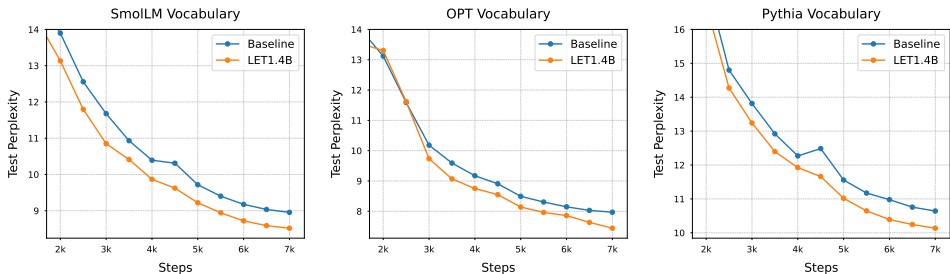

Figure 2: Language modeling performance of LET across three different vocabulary settings. We evaluate the perplexity of models trained with different vocabulary: SmolLM, OPT, and Pythia. For fair comparison (Gao et al., 2020), each subplot uses the same vocabulary. The results demonstrate that LET consistently achieves lower perplexity across all three settings.

**LET Improves Language Modeling**   To comprehensively assess the effectiveness of LET, we evaluate not only the average performance on nine downstream tasks, but also the language modeling perplexity on the test split of The Pile (Gao et al., 2020). For representation alignment in the 1.4B target model $\mathcal{M}$, we use three distinct small pretrained models $\mathcal{T}$ at approximately 125–160M scale (OPT-125M, Pythia-160M, and SmolLM-135M). As shown in Figure 2, each subfigure uses a consistent vocabulary across $\mathcal{M}$ and $\mathcal{T}$. Despite using different small pretrained models, LET consistently reduces test perplexity, in line with the performance improvements observed in downstream tasks. This confirms the robustness of LET in enhancing modeling capability, irrespective of the tokenization scheme employed. Moreover, different small pretrained models have varying impacts: although their sizes are similar, substantial differences in architecture (see Appendix F) lead to different learned representations and, consequently, distinct training dynamics in $\mathcal{M}$. Among these, using SmolLM as $\mathcal{T}$ yields the best overall performance.

**LET Accelerates Training** In addition to improving performance, LET also significantly accelerates training. As shown in Table 1, LET attains higher performance while requiring less than two-thirds of the training steps needed to surpass the baseline. This represents a substantial speedup, even when $\mathcal{T}$ is an order of magnitude ($10\times$) smaller than $\mathcal{M}$. A similar pattern is observed in Figure 2, where LET achieves lower test perplexity during training across three different vocabularies. These results demonstrate LET's effectiveness in accelerating convergence for both language modeling and generalization across diverse tasks. Furthermore, LET not only facilitates efficient pretraining for LLMs, but also exhibits strong cross-domain performance. For additional results on cross-domain generalization, such as time series classification, please refer to Section E.

## 3.3 ABLATION STUDY AND ANALYSIS

**More diverse layer-wise alignment experiments** In our proposed *late-to-early-layer learning* paradigm, we use the late-layer representations of a small pretrained model $\mathcal{T}$ to align the earlier-layer representations of the target model $\mathcal{M}$ during training. This approach has shown strong empirical performance. To systematically assess the impact of different alignment strategies, we conduct a series of ablation experiments with diverse layer alignment configurations. Specifically, we consider six variants: **L2E**, **L2M**, **L2L**, where the last layer of $\mathcal{T}$ aligns with the early, middle, or last layer of $\mathcal{M}$, respectively; and **M2E**, **M2M**, **M2L**, where a middle layer of $\mathcal{T}$ aligns with the early, middle, or last layer of $\mathcal{M}$, respectively. In both the 1B scale and 7B scale settings.

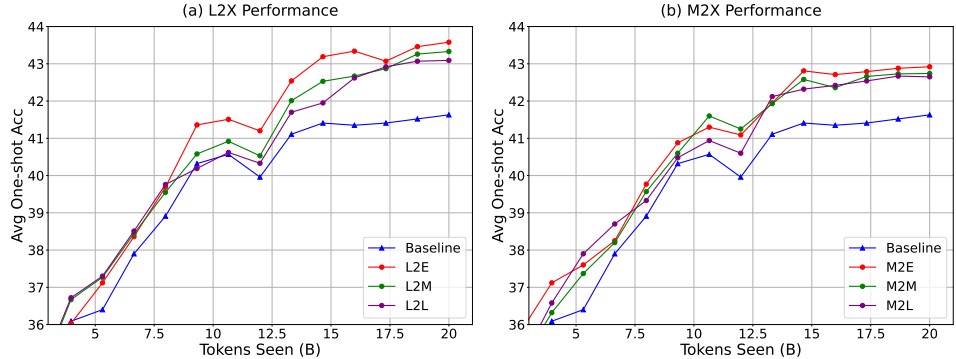

Figure 3: Comparison of six layer-wise alignment strategies on average downstream task performance in one-shot evaluation. The proposed LET paradigm, corresponding to L2E, achieves the highest average performance across all downstream tasks, outperforming all alternative strategies.

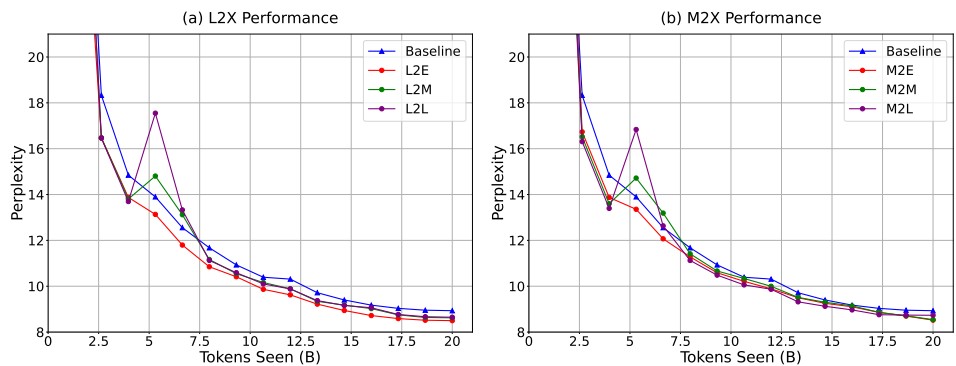

Figure 4: Comparison of six layer-wise alignment strategies on language modeling performance, measured as test perplexity on the test split of The Pile dataset. Both M2E and L2E maintain robust performance throughout training, with L2E yielding the lowest final perplexity among all strategies.

From Figure 3, we can draw two main observations. First, using the middle-layer representations of $\mathcal{T}$ for alignment consistently yields weaker performance compared to using the final-layer representations, as evidenced by M2E, M2M, and M2L underperforming all of L2E, L2M, and L2L. Second,

among all configurations that use the late layer of $\mathcal{T}$ for alignment, L2E demonstrates superior performance and robustness.

As illustrated in Figure 4, the L2E alignment strategy demonstrates superior robustness compared to alternative approaches. This stability is evident from the perplexity trajectories observed after the alignment phase: while all non-L2E strategies show varying degrees of perplexity increase immediately post-alignment, L2E maintains consistent performance. This robustness advantage suggests a more seamless integration between the alignment objective and the underlying language modeling capability. Moreover, L2E achieves the lowest perplexity among all approaches and correspondingly delivers the highest average performance across downstream tasks, further indicating its effectiveness as an alignment strategy.

These empirical results further validate the effectiveness of our late-to-early-layer learning design, in which late-layer representations from the small pretrained model $\mathcal{T}$ guide the formation of informative early-layer representations within the target model $\mathcal{M}$. Consistent with the design rationale of LET, the robustness of L2E can be attributed to its alignment strategy: by mapping the representations of $\mathcal{T}$ to the early layers of $\mathcal{M}$, the subsequent layers retain sufficient capacity to adapt and refine these representations through the learning dynamics of training. This structural configuration becomes increasingly important as training progresses, since $\mathcal{M}$ gradually gains capability and may eventually surpass $\mathcal{T}$ in overall performance, thereby diminishing the relative strength of the alignment representations from $\mathcal{T}$. The remaining layers after early alignment act as a buffer, enabling the seamless integration and progressive refinement of representations provided by $\mathcal{T}$. The trends observed in Figure 4 provide further empirical support for this explanation.

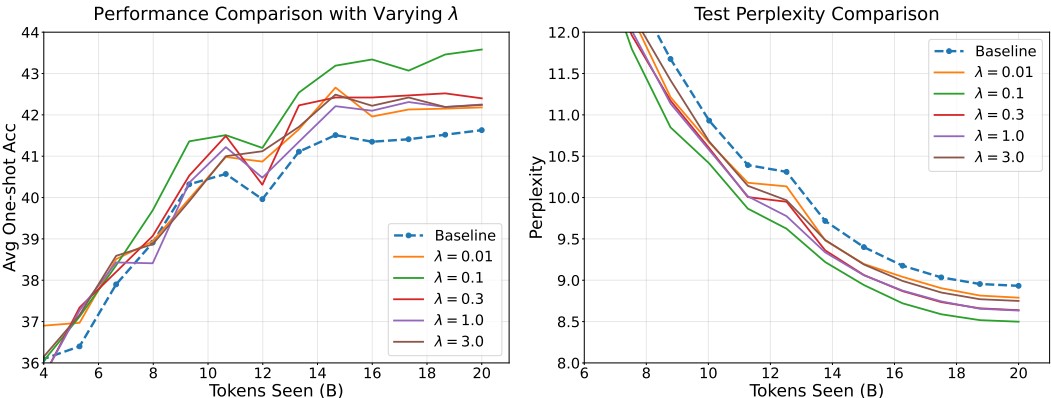

Figure 5: Average downstream task performance (left) and test perplexity on the The Pile dataset (right) evaluated under different $\lambda$ values: 0.01, 0.1, 0.3, 1.0, and 3.0. "Baseline" denotes training with standard causal language modeling, whereas all other configurations employ the proposed LET paradigm with different $\lambda$.

**Effect of hyperparameter $\lambda$ on performance** In previous experiments, we adopted $\lambda = 0.1$ as the default setting. To further investigate the effect of the hyperparameter, we conducted additional evaluations across multiple values, specifically $\lambda \in \{0.01, 0.1, 0.3, 1.0, 3.0\}$. The average downstream task performance for each setting is shown in Figure 5. As illustrated, when $\lambda$ exceeds 0.1, performance consistently drops, indicating that larger values induce excessive alignment of the target model $\mathcal{M}$ to the representations of the small pretrained model $\mathcal{T}$ (see Figure 6), which in turn hampers learning from data. Conversely, setting $\lambda = 0.01$ yields performance above the baseline but still below that achieved with $\lambda = 0.1$, suggesting that alignment is insufficient at this lower value and thus limits the effective utilization of representations from $\mathcal{T}$.

Figure 5 and Figure 6 together indicate that $\lambda = 0.1$ achieves an optimal balance, as both excessively large and small values result in suboptimal performance. In addition, Figure 6 reveals the following: (1) higher $\lambda$ values correspond to higher average cosine similarity, reflecting stronger alignment between $\mathcal{M}$ and $\mathcal{T}$; (2) representation similarity increases steadily throughout training, regardless of the $\lambda$ setting; and (3) despite varying $\lambda$ by an order of magnitude, similarity curves remain relatively stable, suggesting that even small $\lambda$ values can provide effective alignment. Overall, $\lambda = 0.1$ offers a

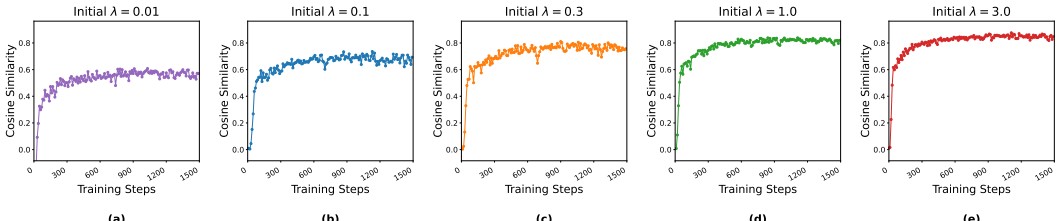

Figure 6: Cosine similarity between the late-layer representations of the small pretrained model $\mathcal{T}$ and the early-layer representations of the target model $\mathcal{M}$ under varying $\lambda$ values.

well-balanced trade-off between aligning with $\mathcal{T}$ and acquiring new knowledge from data, resulting in optimal performance on both downstream tasks and language modeling perplexity.

**LET-1.4B Achieves Superior Performance over Baseline-3B** As shown in Figure 7, LET-1.4B achieves higher performance than Baseline-3B, despite having fewer parameters. This result indicates that the proposed LET paradigm enables the model to learn more effectively from limited training data. LET achieves this by effectively leveraging the representations learned by the model $\mathcal{T}$ to guide alignment in the target model $\mathcal{T}$, thereby improving learning efficiency. This improved efficiency allows models to generalize better with constrained data, making LET particularly valuable in resource-limited settings.

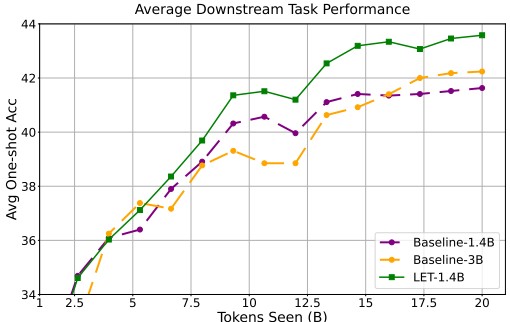

Figure 7: A comparison of average downstream task performance across different training paradigms and model sizes.

## 4 DISCUSSION

In this section, we discuss the potential limitations and future directions of our work.

**Limitations** First, in order to ensure a fair comparison of wall-clock training time, we provide a detailed analysis of the throughput for each method. Due to space constraints, the detailed throughput results are provided in Table 4 in the Appendix F. As shown in the table, the throughput of our LET approach is slightly lower than that of the baseline. Second, while our extensive experiments provide strong evidence for the effectiveness and efficiency of the proposed LET paradigm, empirical evaluations are primarily conducted on models with 1.4B, 3B, and 7B parameters, trained on datasets comprising up to 20B tokens, due to computational resource limitations. Further scaling of both model size and training data is needed to fully demonstrate the scalability of the LET paradigm.

**Future Work and Discussion** First, LET is only applied during the early stages of training. As training progresses and more data is processed, the computational overhead introduced by LET becomes increasingly negligible. Additionally, although RKD achieves marginally higher throughput, its final performance remains substantially inferior to that of LET. Notably, although the baseline achieves $1.078\times$ the throughput of LET during the early training phase, our LET paradigm attains a $1.6\times$ speedup in convergence, which more than compensates for the modest reduction in throughput. Furthermore, when scaling from a 1.4B model to a 7B model, the size of the small teacher model increases by more than an order of magnitude (from SmolLM-135M to SmolLM-1.7B), yet the resulting decrease in throughput remains minimal. This demonstrates that LET is not only efficient but also highly scalable. Second, further validation on larger models, such as those with 70B parameters or more and on substantially larger datasets (e.g., datasets containing 1T tokens) is warranted for thoroughly assessing the practical applicability of LET in real-world settings.

## 5 CONCLUSION

This paper presents Late-to-Early Training (**LET**), a novel paradigm that transforms the vast computational investments already made by the community into a driving force for building stronger LLMs, ensuring that these expensive resources are maximally utilized. Unlike conventional knowledge distillation, which typically relies on substantially larger teacher models, thereby incurring significant memory overhead and may not enable the student to outperform its teacher, LET can exploit much smaller pretrained models to iteratively enhance the capabilities of larger target models. LET introduces two core mechanisms late-to-early-step learning and late-to-early-layer learning, which achieve faster convergence and superior performance without imposing architectural constraints. Extensive experiments across models with 1.4B to 7B parameters validate the effectiveness of LET. Overall, LET offers a pratical pathway for advancing next-generation LLMs, guiding language model development toward a more resource efficient trajectory.

## ETHICS STATEMENT

This work focuses on accelerating the training of LLMs. Although gains in training efficiency may have broader societal implications, we think none of them must be specifically discussed here.

## REPRODUCIBILITY STATEMENT

We are committed to the reproducibility of this work. All models and datasets used are publicly available. Section 3.1 and Appendix B provide a complete description of the experimental setup, including model architectures, training hyperparameters. To further facilitate verification, our source code for training, and evaluation will be made publicly available upon publication.

## ACKNOWLEDGEMENTS

This work was supported by the National Natural Science Foundation of China under Grant No. 62506317 and Doubao Large Model Fund, ByteDance.

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

## A    STATEMENT ON THE USE OF LLMS

In preparing this manuscript, we employed LLMs for linguistic refinement, including identifying and correcting typographical errors and minor grammatical issues, as well as rephrasing sentences to improve clarity and overall readability. LLMs were not involved in the formulation of research ideas, methodological design, experimental execution, data analysis, or the interpretation of results.

## B    EXPERIMENTAL SETTINGS AND DETAILS

This section provides comprehensive experimental settings and implementation details to facilitate full reproducibility of our work.

**Training Hyperparameters** For the experimental configuration, we employed a total batch size of 2048 with an input sequence length of 1280 tokens, resulting in approximately 2.62 million tokens per step. We used The Pile dataset with all copyrighted content removed and define the third layer of $\mathcal{M}$ as the early layer in all configurations. The training setup was adapted based on model size. For the 1.4B parameter model, we utilized a per-GPU batch size of 16 with a gradient accumulation factor of 4. In contrast, for the larger 7B parameter model, we reduced the per-GPU batch size to 4 while increasing gradient accumulation to 16 to accommodate memory constraints. Both models shared common hyperparameters: we applied the AdamW optimizer with $\beta_1 = 0.9$ and $\beta_2 = 0.999$, a weight decay of $0.01$, and a maximum gradient norm of $1.0$ for gradient clipping. The learning rate varied by model size: $4 \times 10^{-4}$ for the 1.4B parameter model and $3 \times 10^{-4}$ for the 7B parameter model, with both utilizing a cosine learning rate schedule for optimization.

**Evaluation and Benchmark** For model evaluation, we assessed our models using nine downstream tasks (used in OLMo). The task suite includes Hellaswag (Zellers et al., 2019), Winograde (Levesque et al., 2012), LAMBADA (Paperno et al., 2016), OpenbookQA (Mihaylov et al., 2018), ARC-easy/challenge (Clark et al., 2018), PIQA (Bisk et al., 2020), SciQ (Welbl et al., 2017), BoolQ (Clark et al., 2019). Regarding perplexity measurements on The Pile dataset, we conducted evaluations at regular intervals of 500 training steps, corresponding to approximately 1.3 billion tokens of training. For downstream task assessments, we saved checkpoints throughout the training process and evaluated them using the EleutherAI evaluation harness framework (Gao et al., 2021). To optimize evaluation efficiency, we use automatic batch size detection within the evaluation harness to identify the maximum supported batch size for each model configuration. Consistent with our training setup, all evaluations were performed on NVIDIA A100 80GB GPUs.

**Baseline Setup** For RKD and SALT experiments, we follow the settings from Rawat et al. (2024). Unless otherwise specified, we use SmolLM2 (referred to as SmolLM for brevity) as the small model in this paper, with SmolLM-135M for the 1.4B model and SmolLM-1.7B for the 7B model.

## C    RELATED WORK

In this section, we review existing works relevant to LET in details.

### C.1    KNOWLEDGE TRANSFER

**Traditional knowledge distillation and its variants** Traditional knowledge distillation (KD) (Hinton et al., 2015; Romero et al., 2014) involves transferring knowledge from a larger, well-trained teacher model to a smaller student model by minimizing the difference between their output distributions. KD methodologies can be systematically categorized into two principal approaches: logits-based and hint-based techniques. The former operates at the level of output logits. Conversely, hint-based methodologies focus on aligning intermediate representations. Model including DistillBERT (Sanh et al., 2019), DistillBiLSTM (Tang et al., 2019), MINILLM (Gu et al., 2023), MiniMA (Zhang et al., 2023) and MixKD (Liang et al., 2020) adhere to the logits-based distillation paradigm. In contrast, models such as TinyBERT (Jiao et al., 2019), MobileBERT (Sun et al., 2020), MiniLM (Wang et al., 2020), TED (Liang et al., 2023), MetaDistil (Zhou et al., 2021), and AD-KD (Wu et al., 2023b) implement hint-based techniques to establish correspondence between intermediate representations of the teacher and student models. In the domain of computer vision,

Touvron et al. (2021) achieved competitive results by having the student learn from the teacher through attention. To address training efficiency, He et al. (2022) introduced the KDEP framework for efficient pre-training by aligning feature. Chen et al. (2022) proposed a two-stage approach to improve data efficiency. Recent work has advanced the theoretical understanding of KD by demonstrating that the projector enables relational gradients for the student model (Miles & Mikolajczyk, 2024). In parallel, orthogonal projection has proven highly effective, yielding significant enhancements in object detection and image generation (Miles et al., 2024). Most KD approaches focus on scenarios where the teacher model is larger than the student model. In contrast, our work investigates the reverse setting. This setup provides a pathway toward developing next-generation models that aim to balance strong performance with improved memory efficiency and throughput.

**Weak to strong** The idea that weaker models can enhance stronger ones has been explored in various forms, with the concept of weak-to-strong generalization formalized by Burns et al. (2023). They demonstrated that fine-tuning a strong pretrained model on labels generated by a weaker model consistently yields performance surpassing that of the weak supervisor, terming this phenomenon weak-to-strong generalization. Earlier work laid the groundwork for this concept. For instance, work like (Furlanello et al., 2018) showed that in computer vision and language modeling tasks, student models can outperform equivalently sized teacher models without requiring a larger teacher, suggesting inherent robustness in knowledge transfer. MagicDistillation (Shao et al., 2025) introduces a weak to strong distillation framework that enables few step inference for large-scale video diffusion models. In language modeling (Qin et al., 2021; Lee et al., 2023; Rawat et al., 2024) has highlighted the potential and limitations of leveraging weaker models to assist the training of larger models. Within the Mix of Experts (MoE) paradigm, various studies have undertaken significant explorations (He et al., 2024; Liew et al., 2025). Notably, Liew et al. (2025) advances our understanding by identifying empirical scaling laws that characterize the relationship between performance and both dataset size and model configuration. While these works provide valuable insights, they often focus on small-size models (fewer than 1B parameters) or settings where the student is only marginally larger than the teacher with similar architecture. In contrast, our study involves architecture-agnostic student models scaling up to 7B parameters, with the student can be up to $10\times$ larger than the teacher. Moreover, we focus on reusing released open-source models in the community. These models have consumed significant computational resources during their initial pretraining, yet are often underutilized when training new models. Our approach aims to leverage these existing assets more effectively, providing a resource-efficient path for improving larger models using smaller, accessible ones.

## C.2 TRAINING ACCELERATION METHODS

Two lines of research have been particularly active in accelerating language model training: data selection and model growth. Data selection aims to improve training efficiency by improving the quality and diversity of the data used during pretraining (Lin et al., 2024; Li et al., 2023; Liu et al., 2023b). Recent progress has been made in both offline and online selection strategies. Offline methods (Yang et al., 2022; Xie et al., 2023a; Tirumala et al., 2023; Xia et al., 2024; Xie et al., 2023b; Gao et al., 2025) typically involve pre-filtering or reweighting data before training, whereas online methods (Lin et al., 2024; Xia et al., 2023; Chen et al., 2023) dynamically adjust the data distribution during training. Recent work (Gu et al., 2024) revisits data selection from the perspective of optimal control, offering new theoretical insights into selection dynamics. Model growth, initially explored in the 1990s (Fahlman & Lebiere, 1989; Fahlman, 1990; Gutstein et al., 2008), was significantly advanced by Net2Net (Chen et al., 2015), which introduced function-preserving expansions along both the width and depth dimensions. This paradigm has been extended in several directions. Bert2Bert (Chen et al., 2021), Lemon (Wang et al., 2023b), StackedBERT (Gong et al., 2019), LiGO (Wang et al., 2023a) and other related methods (Du et al., 2024; Samragh et al., 2024) focuses on width expansion, depth expansion, or learning-based mapping.

Learning dynamics has provided a value perspective on studying how optimizers select minima (Jastrzkebski et al., 2017; Li et al., 2017; Wu et al., 2018; Xu et al., 2018; Hu et al., 2019; Nguyen et al., 2019; Zhu et al., 2019; Xie et al., 2020; Li et al., 2021). This line of research also suggests that learning dynamics critically affect convergence speed and matters to final minima selection (An, 1996; Neelakantan et al., 2015; Zhou et al., 2019; Xie et al., 2021b; HaoChen et al., 2021; Xie et al., 2023c; Tang et al., 2025). However, relevant methods (Xie et al., 2021c;a; 2022; 2023d) made

great success on training relatively smaller models, such as ResNet, they failed to propose practical training algorithms that work well in LLM training.

These methods effectively improve model convergence efficiency, but they still require carefully designed depth and width expansion strategies, which increase the overall complexity, particularly given the growing number of attention variants and the potential need for additional data pre-processing or complex online selection strategies. LET is orthogonal to these approaches and instead leverages a small pretrained model to accelerate the early stage of model training.

## D SUPPLEMENTARY EMPIRICAL RESULTS

This section provides supplementary empirical results on LLMs, including additional 1B-scale results, detailed comparisons with RKD, analysis of stopping thresholds, and experiments using LLaMA 3.2 1B as the model $\mathcal{T}$.

**Additional experiments with 1B-scale models**  In Section 3.2, we presented results using SmolLM-135M (Allal et al., 2025) as model $\mathcal{T}$. Here, we further extend our investigation by pre-training a 1.4B model with OPT-125M Zhang et al. (2022) and Pythia-160M (Biderman et al., 2023) as the models $\mathcal{T}$, following the same experimental setup described previously 3.1. As shown in Figure 8, despite the $\mathcal{T}$ being significantly smaller than the target model and differing in architecture, we still observe substantial improvements and faster convergence. These results highlight the robustness of our proposed L2E paradigm, which consistently delivers strong performance across different choices of models $\mathcal{T}$.

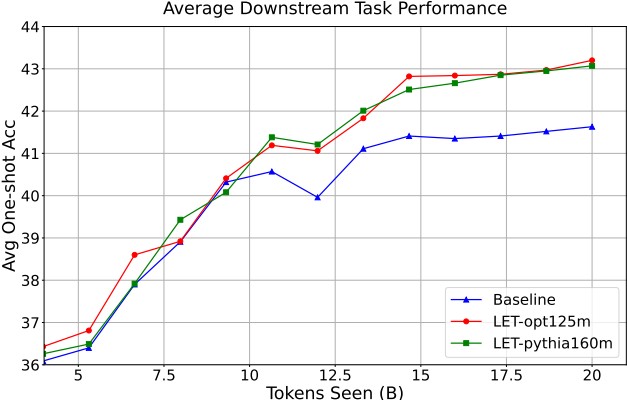

Figure 8: A comparison of average downstream task performance when using Pythia-160M and OPT-125M as the model $\mathcal{T}$. Here, "LET-opt125m" and "LET-pythia160m" represent the use of the LET paradigm with OPT-125M and Pythia-160M as the small models $\mathcal{T}$ respectively.

**Downstream task comparisons with RKD**  In Section 3.2, we compared the final average downstream task performance of RKD with the baseline and our L2E paradigm. In this section, we provide further insight by examining how the average downstream task performance evolves during training when using the RKD. As shown in Figure 9, RKD consistently underperforms compared to the baseline on both 1.4B and 7B models. This observation aligns with the findings of Lee et al. (2023); Rawat et al. (2024). The former, based on experiments with 67M-size models, found that knowledge distillation can degrade performance when the teacher model is at least 0.78 times smaller than the student model. The latter primarily focused on 1.5B and 2.8B models and similarly observed that RKD underperforms the baseline. Moreover, we also observe that the performance degradation of the RKD method is more pronounced on the 7B scale compared to the 1.4B scale.

These results further underscore the performance advantage of our proposed L2E paradigm, which achieves up to $1.6\times$ speedup and a 5.13% improvement in performance even when the model $\mathcal{T}$ is $10\times$ smaller than the target model $\mathcal{M}$. While the work (Lee et al., 2023; Rawat et al., 2024) was highly valuable and provided inspiration for subsequent research, we believe that as language

models become increasingly powerful and are trained on ever-growing datasets, even much smaller models $\mathcal{T}$ can still provide useful guidance during the early stages of training. The results in Figure 9 support this analysis.

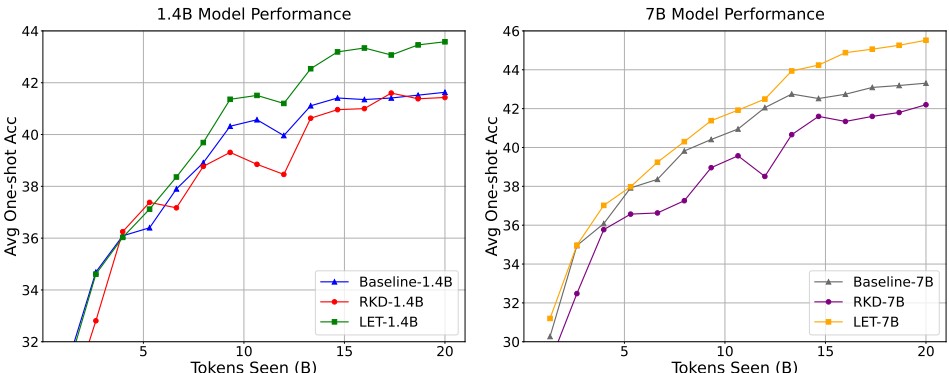

Figure 9: A comparison of average downstream task performance between RKD, Baseline, and LET paradigm at both 1.4B and 7B model scales. We used SmolLM-135M and SmolLM-1.7B as the models $\mathcal{T}$, respectively.

**Experiments and analysis of Different $S_{\mathbf{stop}}$ Values**    To gain preliminary insights into the choice of $S_{\text{stop}}$, we conducted experiments by setting $S_{\text{stop}}$ to 1500 and 3000, respectively. As shown in Figure 10, when training reaches around 5B tokens, using $S_{\text{stop}} = 3000$ yields better performance. This can be attributed to the gradually decreasing $\lambda$ schedule described in Section 2: with a larger $S_{\text{stop}}$, the alignment strength remains higher for a longer period during the early stages of training, which is beneficial for initial learning. However, as training progresses and the student model, being much larger, develops a greater capacity to capture complex knowledge, continued alignment with a much smaller teacher model can actually hinder further improvement. The results in Figure 10 support this analysis.

Ultimately, we choose $S_{\text{stop}} = 1500$, which yields better final performance while reducing overall training time. For a more detailed discussion on wall-clock training time, please refer to Section 4.

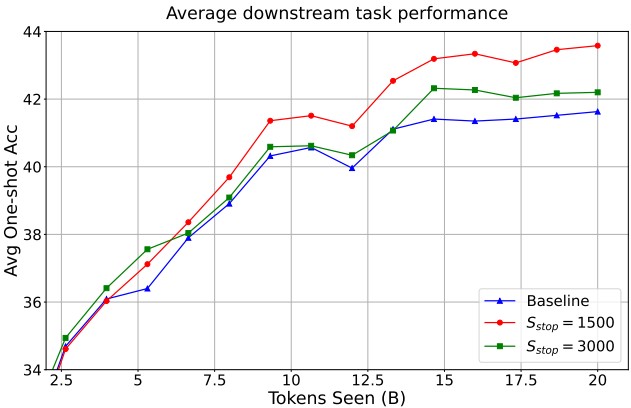

Figure 10: A comparison of average downstream task performance using different stopping thresholds in the LET paradigm. In this experiment, $S_{stop} = 1500$ and $S_{stop} = 3000$ represent implementations of the LET paradigm where alignment was terminated after 1500 and 3000 steps respectively.

**LLaMA 3.2 1B as the model $\mathcal{T}$ for 7B-scale model**    We presented results on the 7B model using SmolLM-1.7B in Section 2. To enable a broader empirical analysis and further evaluate the generalizability of our LET paradigm on 7B models, we conduct additional experiments in this section using Llama-3.2-1B as the model $\mathcal{T}$.

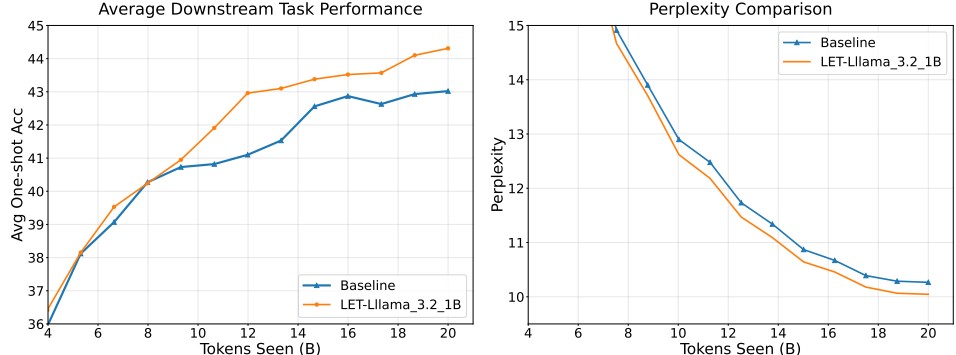

Figure 11: Average performance of the 7B model on downstream tasks (left) and perplexity on test split of The Pile dataset (right). The `LET-Llama_3.2_1B` model is trained using our proposed LET paradigm and leverages Llama-3.2-1B as the model $\mathcal{T}$. In contrast, the baselines are trained using standard causal language modeling. Both models have 7B parameters and share the Llama-3.2-1B vocabulary.

Table 2: Comparison of accuracy between baseline (Qwen-0.5B) and LET across various datasets. Our experimental setup is to add alignment losses at a depth of 6 (24 in total).

| Dataset | Qwen-0.5B | LET (Ours) |
|---|---|---|
| EthanolConcentration | 25.8% | **28.8%** |
| FaceDetection | 59.2% | **66.9%** |
| Handwriting | 23.0% | **33.2%** |
| HreatBeat | 66.8% | **75.0%** |
| JapaneseVowels | 83.9% | **95.7%** |
| PEMS-SF | 45.5% | **62.5%** |
| SelfRegulationSCP1 | 69.6% | **85.5%** |
| SelfRegulationSCP2 | 50.0% | **53.9%** |
| SpokenArabicDigits | 99.3% | **99.7%** |
| UWaveGestureLibrary | 67.5% | **82.2%** |

As shown in Figure 11, applying the LET paradigm to the 7B model also yields significant acceleration and noticeable improvements in final performance. Although the final performance gain is smaller than that observed when using SmolLM-1.7B, the acceleration ratio remains similarly high. We attribute this to the larger parameter size of SmolLM-1.7B, which enables stronger language modeling capabilities and thus provides more effective alignment representations.

**Comparison with SALT**  Table 1 presents a comparison between LET and SALT under identical hyperparameter configurations. SALT employs a two-stage training paradigm controlled by a hyperparameter $n_{\text{KD}}$: KD is applied for the first stage, followed by standard training. For fair comparison, we set $n_{\text{KD}} = S_{\text{stop}}$ to align with our experimental configuration. Our empirical results demonstrate that LET achieves superior performance within the same token budget and LET exhibits stable training dynamics. Additionally, SALT optimizes a different training objective from ours (UL2 (Tay et al., 2022)), and its default hyperparameters may therefore be suboptimal for our training protocol. Given our limited computational budget, we did not conduct a dedicated hyperparameter search for SALT in this setting.

In summary, our extensive empirical analyses in both Section 3 and this section consistently demonstrate the effectiveness and efficiency of our proposed LET.

## E  TIME SERIES EXPERIMENTS

The applicability of LET extends beyond LLMs. To demonstrate its versatility, we evaluate LET on time series classification tasks. As demonstrated in Table 2, we evaluated LET on a diverse set of time series datasets, including EthanolConcentration, FaceDetection, Handwriting, Heartbeat, JapaneseVowels, PEMS-SF, SelfRegulationSCP1, SelfRegulationSCP2, SpokenArabicDigits,

and UWaveGestureLibrary (Wang et al., 2024). The results indicate that LET significantly outperforms the baseline, which involved fine-tuning Qwen-0.5B on the respective tasks. Furthermore, the model $\mathcal{T}$ is the TimesNet (Wu et al., 2023a), specifically pre-trained on a subset of these time series datasets (EthanolConcentration, FaceDetection, Handwriting, Heartbeat, JapaneseVowels, SelfRegulationSCP1, and UWaveGestureLibrary). These empirical findings strongly validate both the generalizability and effectiveness of LET.

# F  LM ARCHITECTURE AND THROUGHPUT

In this section, we detail the model configurations and training efficiency of our experiments.

Table 3: LM architecture comparison

|  | Hidden size | Intermediate size | Num layers | Num heads | Activation | Attention variant |
|---|---|---|---|---|---|---|
| **1B scale setting** | | | | | | |
| OPT-125M | 768 | 3072 | 12 | 12 | Relu | Full |
| Pythia-160M | 768 | 3072 | 12 | 12 | Gelu | Full |
| SmolLM2-135M | 576 | 1536 | 30 | 9 | Silu | GQA |
| Ours-1B | 2048 | 5461 | 24 | 32 | SwiGLU | Full |
| **7B scale setting** | | | | | | |
| SmolLM2-1.7B | 2048 | 8192 | 24 | 32 | Silu | GQA |
| Llama3.2-1B | 2048 | 8192 | 16 | 32 | Silu | GQA |
| Ours-7B | 4096 | 11008 | 32 | 32 | SwiGLU | Full |

Table 4: Training Efficiency and Resource Consumption Comparison. Throughput Ratio is defined as the throughput of the corresponding method divided by the baseline throughput. Wall-Clock Ratio and Peak-VRAM Ratio are defined in the same way.

| Method | Throughput (token/s) | Throughput Ratio ↑ | Wall-Clock Ratio ↓ | Peak-VRAM Ratio ↓ |
|---|---|---|---|---|
| 1.4B Model | | | | |
| Baseline | 224.2k | 1.000 | 1.000 | 1.000 |
| RKD | 211.1k | 0.9415 | 1.0621 | 1.1742 |
| SALT | 221.5k | 0.9880 | 1.0122 | 1.1742 |
| LET | 220.8k | 0.9848 | 1.0154 | 1.1544 |
| 7B Model | | | | |
| Baseline | 105.9k | 1.000 | 1.000 | 1.000 |
| RKD | 98.3k | 0.9282 | 1.0773 | 1.0946 |
| SALT | 104.3k | 0.9849 | 1.0153 | 1.0946 |
| LET | 104.2k | 0.9839 | 1.0163 | 1.0944 |

Table 3 summarizes the architectural configurations of the models used in our empirical analysis. Remarkably, the LET paradigm achieves significant improvements despite substantial architectural heterogeneity among these models. The differences span several dimensions, including hidden size, intermediate size, number of layers, number of attention heads, activation functions, and attention mechanisms. For example, activation functions vary across models, including ReLU (Nair & Hinton, 2010), GeLU (Hendrycks & Gimpel, 2016), SiLU (Elfwing et al., 2018), and SwiGLU (Shazeer, 2020). Similarly, the attention variants include `Full`, which denotes standard multi-head attention (Vaswani et al., 2017), and `GQA` referring to Grouped Query Attention (Ainslie et al., 2023).

As shown in Table 4, we compare throughput, wall-clock time, and peak VRAM across methods. Notably, LET achieves lower peak VRAM than other methods requiring auxiliary models when training with large batch sizes. This efficiency stems from LET's focus on learning representations in

$\mathcal{T}$ rather than the larger logit space, thereby reducing memory overhead. It is worth noting that both LET and SALT only require auxiliary models during the early training phase, resulting in minimal impact on wall-clock time and throughput compared to the baseline. While LET exhibits slightly higher wall-clock time than SALT, its lower peak VRAM under large batch training demonstrates considerable potential for scaling.

## G   Hidden States Alignment

In this section, we provide a detailed description of the projection component in section 2.

In our LET framework, the hidden states extracted from the model $\mathcal{M}$ and the model $\mathcal{T}$ may differ in their hidden dimensionality. Specifically, let $h_{\mathcal{M}}^{(k)} \in \mathbb{R}^{B \times S \times d_{\mathcal{M}}}$ and $h_{\mathcal{T}}^{(L)} \in \mathbb{R}^{B \times S \times d_{\mathcal{T}}}$ denote the hidden representations at layer $k$ of the $\mathcal{M}$ and the final layer $L$ of the $\mathcal{T}$, respectively, where $B$ is the batch size, $S$ is the sequence length, and $d_{\mathcal{M}}, d_{\mathcal{T}}$ are the hidden dimensions.

When $d_{\mathcal{M}} \neq d_{\mathcal{T}}$, we apply an projection operation to the hidden state $h_{\mathcal{M}}^{(k)}$ to align dimension with that of the $\mathcal{T}$. Concretely, we apply linear interpolation along the hidden dimension for each token position independently. That is, for each token index $i \in \{1, \ldots, S\}$ and each sample in the batch, the student representation vector $h_{\mathcal{M},i}^{(k)} \in \mathbb{R}^{d_{\mathcal{M}}}$ is interpolated to produce $\tilde{h}_{\mathcal{M},i}^{(k)} \in \mathbb{R}^{d_{\mathcal{T}}}$. This operation treats the hidden dimension as a 1D information. The interpolation formula for each interpolated coordinate $j \in \{0, \ldots, d_{\mathcal{T}} - 1\}$ is:

$$\tilde{h}_{\mathcal{M},i,j}^{(k)} = (1 - \beta_j) \cdot h_{\mathcal{M},i,\lfloor u_j \rfloor}^{(k)} + \beta_j \cdot h_{\mathcal{M},i,\lfloor u_j \rfloor + 1}^{(k)}, \tag{6}$$

where the source index $u_j = j \cdot \frac{d_{\mathcal{M}} - 1}{d_{\mathcal{T}} - 1}$ and $\beta_j = u_j - \lfloor u_j \rfloor$. This procedure preserves endpoint alignment. After that, the representations $\tilde{h}_{\mathcal{M}}^{(k)}$ and $h_{\mathcal{T}}^{(L)}$ are normalized and compared using the cosine similarity loss:

$$\mathcal{L}_{\text{proj}} = -\sum_{i=1}^{S} \frac{\tilde{h}_{\mathcal{M},i}^{(k)\top} h_{\mathcal{T},i}^{(L)}}{\|\tilde{h}_{\mathcal{M},i}^{(k)}\| \cdot \|h_{\mathcal{T},i}^{(L)}\|}. \tag{7}$$

This alignment ensures that the cosine similarity loss can be computed, even when the model $\mathcal{M}$ and $\mathcal{T}$ have different hidden dimensions.

## H   LogSum Loss Setting

Our LET design (Section 2) employs cosine similarity as the measure of similarity between the normalized representations of model $\mathcal{M}$ and model $\mathcal{T}$. Here, we investigate alternative alignment objectives to assess potential performance improvements.

Given that models $\mathcal{M}$ and $\mathcal{T}$ exhibit substantial differences in capacity in our setting, we note that the logsum loss demonstrates promising performance when applied to models with significant capacity gaps (Miles & Mikolajczyk, 2024). Motivated by this observation, we investigate the effect of replacing cosine similarity with logsum loss in the LET.

As shown in Table 5, employing logsum loss consistently outperforms the Baseline, RKD, and SALT, and further improves upon LET. We attribute the effectiveness of logsum loss to its tendency to emphasize regions where representations between $\mathcal{T}$ and $\mathcal{M}$ diverge significantly, which provides explicit guidance by directing model $\mathcal{M}$ to prioritize learning features with the largest discrepancies, which may be particularly beneficial for efficiently aligning the larger model $\mathcal{M}$ with the pre-trained smaller model $\mathcal{T}$ during early training stages.

## I   Theoretical Analysis

We provide a theoretical analysis of why LET promotes smoother optimization landscapes compared to non-early layer alignment. To facilitate analytical tractability, we focus on a simplified setting: a deep linear network, where the representation dimension is set to d for both model $\mathcal{M}$ and model $\mathcal{T}$.

Table 5: Comparison of average downstream task performance under 1-shot setting. LET-LogSum denotes LET with logsum loss, LET-CCA indicates LET using Canonical Correlation Analysis (CCA) for representation alignment, and LET$^\dagger$ represents the tokenizer-mismatch setting where the model $\mathcal{T}$ uses OPT tokenizer while target models $\mathcal{M}$ use SmolLM tokenizer.

| Method | Avg. Performance | Relative Gain |
|---|---|---|
| *Comparison Methods* | | |
| Baseline | 41.6 | - |
| RKD | 41.4 | -0.2 |
| SALT | 42.9 | +1.3 |
| *Our Methods* | | |
| LET-LogSum | 43.7 | +2.1 |
| LET-CCA | 42.7 | +1.1 |
| LET$^\dagger$ | 42.3 | +0.7 |
| LET | 43.6 | +2.0 |

## I.1 SETUP

We begin by specifying the notation that will be used in the subsequent analysis and proofs.

Consider a model $\mathcal{M}$ with $L$ layers defined by:

$$h^{(l+1)} = W^{(l)} h^{(l)}, \quad l = 0, 1, \ldots, L - 1 \tag{8}$$

Here, $h^{(0)} = x \in \mathbb{R}^d$ denotes the input, $W^{(l)} \in \mathbb{R}^{d \times d}$ are the weight matrices, and $h^{(L)}$ is the output. We define $\theta^{(l)} = \mathrm{vec}(W^{(l)}) \in \mathbb{R}^{d^2}$ as the vectorized parameters of layer $l$, and $\Theta = (\theta^{(0)\top}, \ldots, \theta^{(L-1)\top})^\top \in \mathbb{R}^{Ld^2}$ as the complete parameter vector.

The total training objective is:

$$\mathcal{L}_{\mathrm{total}}(\Theta) = \mathcal{L}_{\mathrm{NLL}}(\Theta) + \lambda \cdot \mathcal{L}_{\mathrm{proj}}(\Theta) \tag{9}$$

where $\mathcal{L}_{\mathrm{NLL}}$ and $\mathcal{L}_{\mathrm{proj}}$ are defined as in Section 2. Our analysis focuses primarily on $\mathcal{L}_{\mathrm{proj}}$ to explicitly isolate the structural impact of the alignment depth, as the task loss $\mathcal{L}_{\mathrm{NLL}}$ remains a shared component across different settings.

## I.2 HESSIAN STRUCTURE ANALYSIS

We analyze the curvature properties of the loss landscape using the Hessian matrix.

For the alignment loss at layer $k$:

$$\frac{\partial \mathcal{L}_{\mathrm{proj}}}{\partial \theta^{(j)}} = \mathbf{0}, \quad \forall j \geq k. \tag{10}$$

This arises because the representation $h^{(k)}$ depends on the parameters $\{W^{(0)}, \ldots, W^{(k-1)}\}$.

The Hessian of model $\mathcal{M}$

$$H_{\mathrm{proj}} = \frac{\partial^2 \mathcal{L}_{\mathrm{proj}}}{\partial \Theta \, \partial \Theta^\top}$$

exhibits a structured block form

$$H_{\mathrm{proj}} = \begin{pmatrix} H_{\mathrm{proj}}^{(0:k)} & \mathbf{0} \\ \mathbf{0} & \mathbf{0} \end{pmatrix}, \tag{11}$$

where $H_{\mathrm{proj}}^{(0:k)} \in \mathbb{R}^{kd^2 \times kd^2}$ corresponds to parameters in layers $0, \ldots, k-1$. For any $i$ and $j \geq k$,

$$\frac{\partial^2 \mathcal{L}_{\mathrm{proj}}}{\partial \theta^{(i)} \partial \theta^{(j)\top}} = \frac{\partial}{\partial \theta^{(i)}} \left( \frac{\partial \mathcal{L}_{\mathrm{proj}}}{\partial \theta^{(j)}} \right)^\top = \frac{\partial}{\partial \theta^{(i)}} \mathbf{0}^\top = \mathbf{0}, \tag{12}$$

and, by symmetry of the Hessian, blocks with $i \geq k$ are also zero.

### I.3 CURVATURE BOUND VIA FROBENIUS NORM

We employ the Frobenius norm $\|\cdot\|_F$ as a measurable proxy for the curvature magnitude. Recalling that the spectral norm, denoted as $\|\cdot\|_2$, which dictates the Lipschitz smoothness constant, is upper-bounded by the Frobenius norm (i.e., $\|A\|_2 \leq \|A\|_F$), it follows that establishing a tighter bound on the Frobenius norm implicitly constrains the maximal curvature.

For a block matrix

$$M = \begin{pmatrix} A & \mathbf{0} \\ \mathbf{0} & \mathbf{0} \end{pmatrix}, \tag{13}$$

its Frobenius norm of the block matrix is identical to that of the upper-left block:

$$\|M\|_F = \|A\|_F. \tag{14}$$

This follows directly from the definition, since

$$\|M\|_F^2 = \sum_{i,j} |M_{ij}|^2 = \sum_{i,j \in A} |A_{ij}|^2 = \|A\|_F^2. \tag{15}$$

We adopt the simplified deep linear network setting where all layers share the same structure. Let $H^{(i,j)}$ denote the Hessian block corresponding to the second derivatives with respect to $\theta^{(i)}$ and $\theta^{(j)}$. To derive an analytical upper bound, and for analytical tractability, we postulate a uniform bound on the Frobenius norms of all Hessian blocks. Specifically, we assume there exists a constant $C > 0$ such that for all layer pairs $i, j < L$, the Hessian blocks satisfy

$$\|H^{(i,j)}\|_F \leq C. \tag{16}$$

Utilizing the established Hessian block structure together with the block matrix norm property,

$$\|H_{\text{proj}}\|_F^2 = \|H_{\text{proj}}^{(0:k)}\|_F^2 = \sum_{i=0}^{k-1} \sum_{j=0}^{k-1} \|H^{(i,j)}\|_F^2 \leq k^2 C^2, \tag{17}$$

and taking square roots gives the bound.

From the curvature upper bound $\|H_{\text{proj}}(k)\|_F \leq k\,C$, it follows immediately that, for $k_1 < k_2 < L$, the theoretical upper bound on the total curvature for alignment at depth $k_1$ is smaller than that for alignment at depth $k_2$. This indicates that, within our bounding analysis, earlier alignment layers admit smaller upper bounds on curvature than later ones.

In summary, under the simplified deep linear network model and the uniform Hessian block bound assumption, our analysis shows that LET incurs a smaller theoretical upper bound on the additional curvature cost, thereby preserving more of the original optimization landscape than non-early alignment and ultimately yielding a smoother landscape. Extending beyond this simplified setting, we empirically validate in Section 3 that the smooth optimization landscape induced by LET is consistently observed in modern model architectures.

## J FAILURE MODE ANALYSIS AND LAYER SELECTION STRATEGIES

In this section, we investigate scenarios where LET exhibits limitations and examine the impact of layer selection strategies on final performance.

When employing GPT-2 (Radford et al., 2019) as the small model $\mathcal{T}$, LET underperforms the baseline. As shown in Table 6, we evaluate three configurations: LET-GPT2-Small pairs LET with GPT-2 Small as $\mathcal{T}$, LET-GPT2-Medium uses GPT-2 Medium, and RKD employs GPT-2 Small. The results reveal a progressive improvement from RKD to LET-GPT2-Small to LET-GPT2-Medium, though all variants underperform the baseline. We attribute this degradation to the potentially lower quality of GPT-2's training data (with a cutoff of late 2017) compared to modern language models. Consequently, GPT-2's representations fail to provide effective alignment signals. Notably, LET consistently outperforms RKD, demonstrating superior robustness to model quality.

Aligning the final layer of $\mathcal{T}$ with earlier layers of $\mathcal{M}$, specifically the third layer, yields optimal performance gains. As illustrated in Figure 12, we use SmolLM-135M as $\mathcal{T}$ and evaluate different

| Method | Avg. | Δ |
|---|---|---|
| Baseline | 41.6 | – |
| RKD | 39.2 | -2.4 |
| LET-GPT2-Small | 40.7 | -0.9 |
| LET-GPT2-Medium | 41.1 | -0.5 |

Table 6: Average 1-shot performance across downstream tasks when employing GPT-2 variants as the small model $\mathcal{T}$, which was pre-trained on data up to late 2017.

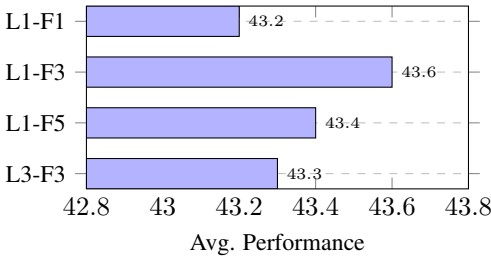

Figure 12: Impact of layer selection strategies

pairing strategies, where L1-F1 denotes aligning the last layer of $\mathcal{T}$ with the first layer of $\mathcal{M}$, with analogous notation for other layer pairs. The results demonstrate that L1-F3 achieves the best performance, which suggests that the first layer may primarily encode input-specific information. The inferior performance of L1-F5 compared to L1-F3 indicates that the third layer strikes an optimal balance for representation alignment.

## K DESCRIPTIONS OF EVALUATION TASKS

We briefly describe each downstream evaluation task used in our experiments, which are intended to help interpret the one-shot performance results reported in the main experiment section 3.

HellaSwag (HS) (Zellers et al., 2019): A commonsense reasoning benchmark where the model needs to choose the most plausible sentence to follow a given context from options. The task is designed to be adversarial against language models through counter-intuitive distractors.

Winogrande (Wino.) (Levesque et al., 2012): A coreference resolution benchmark that evaluates the model's ability to resolve pronouns in sentences requiring commonsense knowledge. It is based on the Winograd schema challenge, scaled up in size and difficulty.

LAMBADA (LAMB) (Paperno et al., 2016): A word prediction task where the model needs to predict the final word of a passage. The passages are filtered to require broad contextual understanding beyond the last sentence.

OpenbookQA (OBQA) (Mihaylov et al., 2018): A multiple-choice question answering task that combines a small "open book" of science facts with commonsense reasoning. The model must integrate both explicit knowledge and inference.

ARC (ARC-c and ARC-e) (Clark et al., 2018): A science question answering benchmark divided into two subsets. The "easy" (ARC-e) set consists of questions that can often be answered with simple reasoning or basic science knowledge, while the "challenge" (ARC-c) set includes more difficult questions requiring complex inference or broader background knowledge.

PIQA (Bisk et al., 2020): A physical common sense reasoning task involving everyday scenarios. The model must select the more plausible solution among candidates for completing an action.

SciQ (Welbl et al., 2017): A science multiple-choice QA dataset with questions crowd-sourced and aligned to middle school science curricula. The task requires a mixture of factual recall and reasoning.

BoolQ (Clark et al., 2019): A binary (yes/no) question answering task over short passages. The model must decide whether the answer to the question is entailed by the given passage.

