# OpenReview forum: "Late-to-Early Training: LET LLMs Learn Earlier, So Faster and Better"
_ICLR.cc/2026/Conference — ICLR 2026 Poster_

### Official Review · Reviewer_srWB · 2025-10-24

**Soundness:** 4
**Presentation:** 3
**Contribution:** 3
**Rating:** 6
**Confidence:** 4

**Summary:**

The authors address the problem of utilizing existing pre-trained smaller models to train larger models. To do this they propose to use KD, which would be an architecture agnostic form of upcycling. Extensive results show improved training using this technique. The main novelty lies in guiding the earlier layers using the late layers of the smaller pre-trained model.

**Strengths:**

The idea of using smaller pre-trained models to accelerate training larger models is a very interesting and relevant problem statement. The proposed approach of using reverse distillation from early to late layers makes a lot of sense and the authors demonstrate is very effective. I don't see any issues with the experimental section or the theory, which is relatively thorough and clear.

The results are shown on large scale models (which is important for this problem statement) and show clear and strong empirical results. The application of KD to this problem is new and interesting.

**Weaknesses:**

Unfortunately, the authors claim of being the first to leverage smaller pre-trained models to train larger LLMs is not true. There is a relatively new technique called upcycling, which has been used extensively in industry and academia [1]. I would really encourage the authors to include a thorough discussion on these works and related techniques. As far as I know, the authors approach for using reverse knowledge distillation is indeed novel, but the problem statement itself is not. Finally, I do understand that the proposed KD technique makes upcycle architecture agnostic and this is a benefit which should be highlighted in this work over prior upcycle techniques.

KD has been shown to be more data efficient than training a model from scratch [2,3,4,5,6]. I would encourage the authors to add this to the discussion and analysis for *why* LET is able accelerate training the larger LLMs.

In summary, I think the paper is missing further analysis with respect to prior works in the KD literature and the novelty of the problem statement is a bit overclaimed. Including a more thorough related work discussion to prior upcycle techniques would be good. If the authors make this these changes I would be very happy to increase my score.

[1] Scaling Laws for Upcycling Mixture-of-Experts Language Models. PMLR 2025

[2] Understanding the Role of the Projector in Knowledge Distillation. AAAI 2024

[3] Training data-efficient image transformers & distillation through attention. PMLR 2021

[4] VkD : Improving Knowledge Distillation using Orthogonal Projections. CVPR 2024

[5] Knowledge Distillation as Efficient Pre-training: Faster Convergence, Higher Data-efficiency, and Better Transferability. CVPR 2022

[6] DearKD: Data-Efficient Early Knowledge Distillation for Vision Transformers. CVPR 2022

**Questions:**

Although only a small modification, it would be interesting to see the results using layer norm and a smooth l1 or logsum loss [2]. These are shown to be more effective for distillation and I am curious if these results extend to the authors proposed setting here. In general, fitting this work into the recent knowledge distillation literature would really strengthen the submission.

---

> ### Author Response · Authors · 2025-11-21
> **Response to Reviewer srWB**
>
> We sincerely appreciate your time in reviewing our paper and providing valuable comments.
>
>
> ## W1: Analysis with respect to prior works
>
> We greatly appreciate the reviewer's suggestion and the valuable references provided. In the revised manuscript, we have expanded our related work section and incorporated the suggested papers. Specifically, we have integrated KD-related discussions into the paragraph "Traditional Knowledge Distillation and Its Variants". We have also added comprehensive discussion of upcycling techniques in the paragraph "Weak-to-Strong Learning".
>
> Our contributions are distinct in several key aspects. First, LET is **architecture-agnostic**, which is increasingly important as the community releases diverse open-source models with varying architectures. Second, our approach remains effective even when the teacher model is 10× smaller than the student, significantly reducing memory overhead and improving training efficiency compared to prior upcycling techniques that typically require similar-sized models. Third, LET provides a smoother landscape (see Figure 4), which is crucial for stable training of larger models.
>
> In the revised manuscript, we have clarified our problem statement to more accurately position LET within the existing body of work, emphasizing our unique contributions while appropriately recognizing relevant prior research.
>
>
> ## W2: Discussion and analysis of why LET accelerates training of larger LLMs
>
> LET introduces two key mechanisms: Late-to-Early Step Learning and Late-to-Early Layer Learning. The latter is particularly crucial. Unlike naive KD that typically aligns model outputs, LET learns structured knowledge at early layers during training. From an ordinary differential equation (ODE) perspective, LET adds intermediate point conditions at early time steps (i.e., early layers), providing structured guidance across the model's hierarchical layers. This is especially important when training a larger model from a smaller one: as training progresses, the larger model may eventually surpass the teacher in overall capacity. In such cases, aligning output logits may produce negative guidance. Our Late-to-Early Layer Learning mechanism mitigates this issue by focusing on early-layer representations rather than final outputs. We empirically validate this advantage in Empirical Analysis Section.
>
> Moreover, with continuous improvements in data quality and training techniques, even small-size open-source models (e.g., Qwen family, OLMo, SmolLM series) now exhibit strong performance. During the early training phase of larger models (before $S_{\text{stop}}$ in our setting), the structured representations learned by these high-quality smaller models can provide positive alignment signals, accelerating convergence.
>
> Finally, one key motivation of this work is to encourage researchers and engineers to fully leverage the computational resources already invested in open-source models. By enabling efficient knowledge transfer from existing models, LET accelerates development of larger, better-performing models while conserving resources.
>
>
> ## Q1: Additional ablation experiments
>
> We sincerely appreciate the reviewer's attention to alignment strategy. LET employs layer normalization before computing cosine similarity. Regarding LogSum Loss, we have conducted additional experiments replacing cosine similarity with logsum loss. As shown in Table 5 (see below), logsum loss consistently outperforms the Baseline, RKD, and SALT, and further improves upon LET. We attribute the effectiveness of logsum loss to its tendency to emphasize regions where representations between $\mathcal{T}$ and $\mathcal{M}$ diverge significantly, which provides explicit guidance by directing model $\mathcal{M}$ to prioritize learning features with the largest discrepancies, which may be particularly beneficial for efficiently aligning the larger model $\mathcal{M}$ with the pre-trained smaller model $\mathcal{T}$ during early training stages.
>
> **Table: Comparison of Average Downstream Task Performance under 1-shot Setting**
>
> *Note: LET-LogSum denotes LET with logsum loss, LET-CCA indicates LET using Canonical Correlation Analysis (CCA) for representation alignment, and LET† represents the tokenizer-mismatch setting where the model $\mathcal{T}$ uses OPT tokenizer while target models $\mathcal{M}$ use SmolLM tokenizer.*
>
> ### Comparison Methods
>
> | Method | Avg. Performance | Relative Gain |
> |:-------|:----------------:|--------------:|
> | Baseline | 41.6 | - |
> | RKD | 41.4 | -0.2 |
> | SALT | 42.9 | +1.3 |
>
> ### Our Methods
>
> | Method | Avg. Performance | Relative Gain |
> |:-------|:----------------:|--------------:|
> | LET-LogSum | 43.7 | +2.1 |
> | LET-CCA | 42.7 | +1.1 |
> | LET† | 42.3 | +0.7 |
> | LET | 43.6 | +2.0 |
>
> We once again thank the reviewer for the constructive feedback.

---

> > ### Comment · Reviewer_srWB · 2025-11-24
> >
> > The new results and improvements are interesting. I believe this consolidates how LET fits into the feature KD literature and how it can easily incorporate ideas from this adjacent field. I have read through the other reviewer comments too and I do agree that a theoretical discussion could be good to see. However, I think a post-hoc explanation (in this case with ordinary differential equations) is not too helpful as it does not inform any of the specific design decisions (or vice versa). This paper is primarily emperical and practically useful, which is fine.
> >
> > The authors have addressed all of my concerns and the updated manuscript has improved a lot. I have updated my confidence and am leaning more towards acceptance. I will read through any other reviewer/author comments throughout the discussion period.

---

### Official Review · Reviewer_dPjE · 2025-10-26

**Soundness:** 3
**Presentation:** 3
**Contribution:** 3
**Rating:** 8
**Confidence:** 3

**Summary:**

Motivated by the desire to re-use smaller open-source LLMs to train larger LLMs efficiently, the authors propose a new method (LET) where a smaller model guides a larger model in order to achieve better performance earlier, reversing the original size relation between student and teacher.  While previous works proposed similar strategies with smaller teachers, the scaling size between the small teacher and the large student has remained small so far. In this work, the authors are able to train 10x larger models by introducing two key modifications: 1) the small teacher is used to align early layer representations of the larger student with its later layers (rather than the logits), 2) this alignment guidance needs to be stronger at the beginning of training and subsides as the larger model gets more capable later in training. The methods is thoroughly tested experimentally.

**Strengths:**

* The writing is very clear
* The experiments are comprehensive with well designed ablation studies
* The reported performance of the method is significant ( "[the method] exceeds the baseline’s average performance while requiring less than 67% of the training steps even with 10× smaller model")
* The method does not require architectural compatibility between the student and the teacher

**Weaknesses:**

* While I am not seeing this as a significant weakness (because of the detailed experimental evidences) the proposed method is lacking theoretical backing.

**Questions:**

* Can the authors comment on theoretical reasons that could underpinne this method.

---

> ### Author Response · Authors · 2025-11-21
> **Response to Reviewer dPjE**
>
> First, we sincerely appreciate the reviewer's supportive evaluation.
>
> We are happy to discuss on the theoretical Insights that could underpin our method from the perspective of Ordinary Differential Equations (ODEs).
>
> Modern LLMs, typically built upon the Transformer architecture and its variants, can be viewed as a discretized ODE where integer time steps correspond to layer indices [1]. The ODE state variables are instantiated as the intermediate hidden representations that progressively evolve across layers, transitioning from input embeddings to final output representations used for next-token prediction.
>
> A crucial concept in solving ODEs is that the solution space can be constrained by applying boundary or intermediate point conditions. Building on extensive research that establishes how both intermediate and final layers of LLMs encode rich, meaningful representations [2, 3, 4], we leverage this insight in our design. Our proposed LET alignment between the target model $\mathcal{M}$’s early representation and the small model $\mathcal{T}$’s final representation effectively imposes an intermediate point condition on the learning trajectory of $\mathcal{M}$. This constraint guides the optimization process, steering the representation evolution of $\mathcal{M}$ towards a semantically meaningful region of the representation space as it learns to fit the training data.
>
> Notably, what sets our LET paradigm apart from traditional KD is its emphasis on transferring abstract knowledge in early state, rather than simply aligning model outputs as is common in conventional KD. This distinction becomes particularly crucial in our unconventional setting of using a smaller model to train a larger one. This is because, as training progresses, the larger model $\mathcal{M}$ is expected to eventually surpass the smaller $\mathcal{T}$ in overall performance. In such settings, persisting with logit-level alignment to $\mathcal{M}$ could introduce negative effects.
>
> However, we know that other theoretical frameworks might also offer valid interpretations. The development of a standard and unified theoretical framework for such alignment techniques in LLMs is still an open and valuable research direction, which we leave for future work.
>
> We extend our gratitude once again for for your recognition of our work's contributions.
>
> [1] Chen, Ricky TQ, et al. "Neural ordinary differential equations." Advances in neural information processing systems 31 (2018).
>
> [2] Dar, Guy, et al. "Analyzing transformers in embedding space." Proceedings of the 61st Annual Meeting of the Association for Computational Linguistics (Volume 1: Long Papers). 2023.
>
> [3] Elhoushi, Mostafa, et al. "Layerskip: Enabling early exit inference and self-speculative decoding." Proceedings of the 62nd Annual Meeting of the Association for Computational Linguistics (Volume 1: Long Papers). 2024.
>
> [4] Skean, Oscar, et al. "Layer by layer: Uncovering hidden representations in language models." arXiv preprint arXiv:2502.02013 (2025).

---

> ### Comment · Reviewer_dPjE · 2025-11-21
>
> This is a very interesting intuition. Thanks for sharing it!
>
> I maintain my score as a good paper to be accepted, while I am raising my confidence level in that judgement given all the improvements made to the paper.

---

### Official Review · Reviewer_ec8P · 2025-11-01

**Soundness:** 2
**Presentation:** 3
**Contribution:** 2
**Rating:** 4
**Confidence:** 4

**Summary:**

The **Late-to-Early Training (LET)** paper proposes a new pre-training method that uses small, existing models to accelerate the training and improve the performance of new, larger models . Its core mechanism involves two parts:

1. **Late-to-Early-Layer (L2E) Learning:** It aligns the internal representations from the **final (late) layer** of the small teacher model with an **early layer** of the large student model .
2. **Late-to-Early-Step (L2S) Learning:** This alignment acts as a temporary guide during the **early training steps** only, fading to zero at a set point (Sstop) .

This L2E + L2S design prevents the small teacher from bottlenecking the larger model, allowing it to serve as a "bootstrap" rather than a "ceiling" . Experiments show this method achieves up to **1.6x faster** convergence while also yielding **higher final accuracy** than standard training, even when the teacher model is 10x smaller .

**Strengths:**

This paper demonstrates that knowledge distillation (KD) onto a large model is possible using a teacher model that is 10x smaller. By applying KD only during the initial phase of pre-training, not every step, the computational cost does not persist throughout the entire training process . The paper presents results showing that this method achieves higher performance compared to standard training without knowledge distillation .

**Weaknesses:**

- **Insufficient Baseline Comparisons:**
The paper compares against **standard training** and **RKD**, but omits head-to-head evaluations with **large-teacher, logits-based KD**, strong **offline KD** pipelines, and recent **data-selection / model-growth** accelerators. Adding **wall-clock–normalized** and **peak-VRAM–normalized** comparisons to these families would more clearly position LET.
- **Size of the L2E Advantage:**

    While Figures 3–4 **suggest** L2E > L2M/L2L, the **visual gaps appear modest**. Please report **exact end-of-training deltas, variance, confidence intervals,** and (where feasible) **repeat runs** to rule out noise. In early steps, L2M/L2L sometimes exceed L2E; clarifying **why early-layer alignment should win eventually** (with theory-backed or empirical ablations) would strengthen the claim.

- **Relation to Offline KD:**

    Prior **offline KD** reports (e.g., MiniPLM, ICLR 2025) claim **2.2×** speedups against their baselines, whereas LET reports **1.6×** against standard training. Unlike LET’s **dual-model forward** in early steps, offline KD typically adds **no per-step training overhead** (though it may incur preprocessing cost). A **fair, wall-clock** comparison—controlling for hardware, batch size, and token budgets—would clarify the net efficiency trade-offs.

- **Training Overhead Transparency:**

    LET requires **co-loading teacher and student** and **forwarding both** during the early phase, introducing **VRAM pressure** and a **throughput hit** (the paper notes roughly **~8%** slower throughput). Please include **end-to-end wall-clock curves**, **tokens/sec**, **peak VRAM**, and **batch-size regressions** across teacher sizes to show that faster convergence outweighs this overhead.

- **Heuristic Dimension Alignment:**

    To resolve teacher–student hidden-size mismatch, the method uses **1-D linear interpolation** of teacher hidden states followed by **cosine-similarity alignment**. The paper should justify what **semantics are preserved** by this interpolation and compare against stronger baselines (e.g., **learned linear projections**, **CCA/Procrustes**, or **adapter heads**) and **tokenizer-mismatch** settings to demonstrate robustness.

**Questions:**

Please refer to the Weaknesses section.

---

> ### Author Response · Authors · 2025-11-21
> **Response to Reviewer ec8P (Part I)**
>
> We sincerely appreciate your time and feedback in reviewing our paper.
>
> ## W1: Insufficient Baseline Comparisons
>
> To provide a more comprehensive evaluation of LET's performance, we have added comparisons with SALT at both the 1B and 7B scales. SALT employs a two-stage training paradigm controlled by a hyperparameter $n_{\text{KD}}$: KD is applied during the first stage, followed by standard training in the second stage. For fair comparison, we set $n_{\text{KD}} = S_{\text{stop}}$ to align with our experimental configuration. Our empirical results demonstrate that LET substantially outperforms not only the baseline and RKD, but also SALT (see Table 1). Notably, compared to RKD and SALT, both of which require an additional teacher model, our method achieves lower peak memory consumption, as detailed in the table below. Note that LET and SALT are not active throughout the entire training process; therefore, we normalize throughput (tokens/s) and wall-clock time globally across all training stages for fair comparison.
>
> **Table: Training Efficiency and Resource Consumption Comparison**
>
> *Note: Throughput Ratio is defined as the throughput of the corresponding method divided by the baseline throughput. Wall-Clock Ratio and Peak-VRAM Ratio are defined in the same way.*
>
> **1.4B Model:**
>
> | Method | Throughput (token/s) | Throughput Ratio ↑ | Wall-Clock Ratio ↓ | Peak-VRAM Ratio ↓ |
> |:-------|---------------------:|-------------------:|-------------------:|------------------:|
> | Baseline | 224.2k | 1.000 | 1.000 | 1.000 |
> | RKD | 211.1k | 0.9415 | 1.0621 | 1.1742 |
> | SALT | 221.5k | 0.9880 | 1.0122 | 1.1742 |
> | LET | 220.8k | 0.9848 | 1.0154 | 1.1544 |
>
> **7B Model:**
>
> | Method | Throughput (token/s) | Throughput Ratio ↑ | Wall-Clock Ratio ↓ | Peak-VRAM Ratio ↓ |
> |:-------|---------------------:|-------------------:|-------------------:|------------------:|
> | Baseline | 105.9k | 1.000 | 1.000 | 1.000 |
> | RKD | 98.3k | 0.9282 | 1.0773 | 1.0946 |
> | SALT | 104.3k | 0.9849 | 1.0153 | 1.0946 |
> | LET | 104.2k | 0.9839 | 1.0163 | 1.0944 |
>
> Regarding the use of a large-teacher model, we clarify that our paper's setting specifically focuses on training a larger model using a much smaller one. In the context of LLMs, employing a substantially larger teacher inevitably incurs considerable memory and computational overhead. Traditional knowledge distillation typically uses a larger, more capable teacher to supervise a smaller student model. However, in such scenarios, student models tend to lag behind their teachers in performance, which limits their utility as a foundation for scaling LLM capabilities. Given these considerations, we deliberately chose a smaller model as $\mathcal{T}$ to demonstrate that effective guidance can be provided without relying on more expensive larger teachers.
>
> Regarding logits-based KD, RKD itself is a classic logits-based distillation approach; thus, our experiments already include comparisons with this family of methods. As for data-selection and model-growth methods, we have discussed them in detail in our related work section (L902-L913). Importantly, LET is orthogonal to these approaches: LET focuses on the training paradigm and is architecture-agnostic, whereas data-selection methods primarily address data curation, and model-growth techniques impose specific architectural constraints.
>
> We appreciate the reviewer's suggestion to include wall-clock time and peak VRAM metrics. We have added this information in Table 4. The results show that LET achieves advantages in peak VRAM compared to other methods requiring additional models. Crucially, LET is only active during the first $S_{\text{stop}}$ steps, after which training proceeds identically to standard training, thereby limiting the additional overhead. While LET exhibits slightly higher wall-clock time than SALT, its lower peak VRAM under large batch training demonstrates considerable potential for scaling.

---

> ### Author Response · Authors · 2025-11-21
> **Response to Reviewer ec8P (Part II)**
>
> ## W2: the Advantage of L2E
>
> We appreciate your concern regarding the statistical significance of our results. In Figures 3 and 4, L2E ultimately achieves superior performance. Regarding exact end-of-training deltas: on perplexity, L2E outperforms L2M and L2L by 0.05 and 0.07, respectively; on average downstream task performance, L2E surpasses L2M and L2L by 0.25 and 0.49, respectively. While obtaining precise variance and confidence intervals would require multiple training runs, we note that reporting such metrics is not standard practice in the pre-training and continual pre-training literature due to the substantial computational cost of each run, see recent work [1][2].
>
> Regarding the observation that L2M and L2L sometimes exceed L2E in early training steps, this behavior is expected and aligns with our framework. In our unconventional setting where a smaller model trains a larger one, the small teacher model, having been trained on substantial data, can provide effective representations to the larger student model during the initial phase. However, as training progresses, the larger model $\mathcal{M}$ is expected to surpass the smaller teacher $\mathcal{T}$ in performance. At this stage, continued alignment with the teacher's outputs (as in RKD) or with the teacher's later-layer representations (as in L2L and L2M) may hinder the student's learning trajectory. Our empirical results demonstrate this effect In Figures 3 and 4.
>
> Furthermore, assessing LLM performance based solely on "extrinsic" evaluation metrics such as perplexity and average downstream task scores may not fully capture a model's capabilities [3][4]. While demonstrating LET's superior final performance in Figures 3 and 4 is important, our equally critical objective is to show that LET exhibits a smoother optimization landscape, which is most evident in Figure 4. This smoother trajectory indicates more stable and efficient learning dynamics, which is a valuable property beyond raw performance gains.
>
> [1] Lin, Zhenghao, et al. "Not all tokens are what you need for pretraining." Advances in Neural Information Processing Systems 37 (2024): 29029-29063.
>
> [2] Gu, Yuxian, et al. "Data Selection via Optimal Control for Language Models." The Thirteenth International Conference on Learning Representations.
>
> [3] Schaeffer, Rylan, Brando Miranda, and Sanmi Koyejo. "Are emergent abilities of large language models a mirage?." Advances in neural information processing systems 36 (2023): 55565-55581.
>
> [4] Wei, Lai, et al. "Diff-erank: A novel rank-based metric for evaluating large language models." Advances in Neural Information Processing Systems 37 (2024): 39501-39521.
>
> ## W3: Relation to Offline KD
>
> We appreciate the reviewer's suggestion to compare with offline KD methods. We would like to clarify that LET and offline KD approaches like MiniPLM address different but complementary scenarios. MiniPLM, employing a larger teacher to train a smaller student model (specifically, a 500M model in their experiments). This conventional setting naturally facilitates their 2.2× speedup. LET, conversely, tackles a different challenge: leveraging a smaller model to accelerate the training of a larger target model (1B and 7B).
>
> Beyond this fundamental difference in problem settings, we note several practical considerations. Offline methods incur substantial preprocessing overhead, making direct comparison challenging. Moreover, as training corpora scale, storing preprocessed teacher outputs becomes increasingly impractical. Finally, while MiniPLM focuses primarily on the Qwen model family, LET demonstrates effectiveness across diverse architectures (SmolLM, OPT, Pythia, and LLaMA), suggesting broader applicability.
>
> ## W4: Training Overhead Transparency
>
> We thank the reviewer for this practical suggestion. In Table 4, we have recorded the throughput (tokens/sec) for different methods and have now added wall-clock time and peak VRAM normalized information. The results demonstrate that among settings requiring an additional teacher model, LET achieves reduced VRAM usage, showcasing its efficiency. Regarding batch size configuration, we have described this in the Experimental Settings and Details section (L825-L828): for the 1.4B parameter model, we utilized a per-GPU batch size of 16 with a gradient accumulation factor of 4; for the 7B parameter model, we reduced the per-GPU batch size to 4 while increasing gradient accumulation to 16 to accommodate memory constraints.
>
> Notably, LET achieves a 1.56$\times$ wall-clock speedup to achieve the same performance. This is attributed to the fact that LET is only active during the first $S_{\text{stop}}$ steps, after which the training overhead matches that of standard training. The choice of using a small model as $\mathcal{T}$ is precisely motivated by the goal of reducing VRAM pressure and improving throughput (tokens/sec).

---

> ### Author Response · Authors · 2025-11-21
> **Response to Reviewer ec8P (Part III)**
>
> ## W5: Alignment Strategy
> We appreciate this valuable suggestion. In designing LET, we deliberately opted for cosine-similarity alignment to maintain LET simplicity while avoiding the additional VRAM pressure and communication overhead that learnable alignment strategies would introduce. Table 1 and Figure 6 validate the effectiveness of this approach: as the hyperparameter $\lambda$ increases, the similarity between $\mathcal{T}$'s representations and $\mathcal{M}$'s representations increases correspondingly. Notably, Table 1 reveals that LET achieves particularly pronounced improvements on ARC-Challenge, Winogrande, and BoolQ, tasks that demand reasoning capabilities. This performance pattern suggests that our early-layer alignment strategy effectively transfers high-level abstract knowledge structures associated with reasoning.
>
> We have also conducted experiments using CCA and LogSum for alignment. As shown in Table 5 (see below), CCA demonstrates competitive performance but does not surpass the cosine-similarity approach used in LET. We attribute this to the fact that during early training, $\mathcal{M}$'s representations may not be sufficiently structured or semantically meaningful, rendering CCA relatively less effective in this regime. Given that models $\mathcal{M}$ and $\mathcal{T}$ exhibit substantial differences in capacity in our setting, we note that the logsum loss demonstrates promising performance when applied to models with significant capacity gaps. Motivated by this observation, we investigate the effect of replacing cosine similarity with logsum loss in the LET. Results indicate that employing logsum loss further improves upon LET. We note that while learned linear projections could be explored, they entail additional memory footprint and inter-device communication overhead. Additionally, during early training phases where $\mathcal{M}$'s representations remain less structured, such learnable approaches may offer limited benefits compared to current design.
>
> **Table: Comparison of Average Downstream Task Performance under 1-shot Setting**
>
> *Note: LET-LogSum denotes LET with logsum loss, LET-CCA indicates LET using Canonical Correlation Analysis (CCA) for representation alignment, and LET† represents the tokenizer-mismatch setting where the model $\mathcal{T}$ uses OPT tokenizer while target models $\mathcal{M}$ use SmolLM tokenizer.*
>
> ### Comparison Methods
>
> | Method | Avg. Performance | Relative Gain |
> |:-------|:----------------:|--------------:|
> | Baseline | 41.6 | - |
> | RKD | 41.4 | -0.2 |
> | SALT | 42.9 | +1.3 |
>
> ### Our Methods
>
> | Method | Avg. Performance | Relative Gain |
> |:-------|:----------------:|--------------:|
> | LET-LogSum | 43.7 | +2.1 |
> | LET-CCA | 42.7 | +1.1 |
> | LET† | 42.3 | +0.7 |
> | LET | 43.6 | +2.0 |
>
> We have also evaluated LET under tokenizer-mismatch settings (see LET†). LET remains effective in these scenarios because it focuses on abstract representations rather than vocabulary-dependent logits. However, performance is lower than in tokenizer-matched settings, as LET was not specifically optimized for cross-tokenizer alignment. We leave this as an interesting direction for future work.

---

> ### Author Response · Authors · 2025-11-27
>
> Dear Reviewer ec8P,
>
> I hope this message finds you well.
>
> As the discussion phase nears its end, we would like to ensure that our responses have sufficiently addressed all your concerns. We remain available to answer any further questions or discuss additional feedback you may have.
>
> We truly value your insights and have incorporated your suggestions to improve our manuscript.
>
> Thank you again for your time and constructive feedback.

---

### Official Review · Reviewer_4pPU · 2025-11-01

**Soundness:** 3
**Presentation:** 3
**Contribution:** 2
**Rating:** 4
**Confidence:** 4

**Summary:**

This paper introduces Late-to-Early Training (LET), which leverages small pre-trained models to accelerate the pre-training of larger  models.
The authors pose the practical question of whether existing small pre-trained models can guide and speed up the early learning of larger target models.

The core idea of LET is to use representations from the late layers of a pre-trained model to guide the early layers of the target model during early training steps.
The method consists of two mechanisms: Late-to-Early Step Learning and Late-to-Early Layer Learning.
These mechanisms aim to accelerate training convergence and improve both language modeling capability and downstream task performance.
The contributions are summarized in three points.
First, the study formulates the previously underexplored problem of generally accelerating the pre-training of much larger LLMs (e.g., 10×) using small pre-trained models.
Second, it proposes the LET paradigm with the two mechanisms above and states that LET is architecture-agnostic.
Third, it provides extensive experiments showing that LET achieves faster training and superior downstream performance compared to standard training.

The experiments evaluate 1.4B- and 7B-parameter models using perplexity on The Pile and accuracy on nine downstream tasks: HellaSwag, WinoGrande, LAMBADA, OpenBookQA, ARC-easy, ARC-challenge, PIQA, SciQ, and BoolQ.
For the 1.4B model, teachers such as OPT-125M, Pythia-160M, and SmolLM-135M yield consistent gains, with up to 1.6× training speedup on The Pile and nearly 5% improvement in downstream accuracy; for the 7B model, using Llama-3.2-1B as the teacher also provides faster training and higher final performance.

**Strengths:**

* The core mechanisms are clear (late-to-early-step / late-to-early-layer).
It formalizes two mechanisms, using a teacher’s late-layer representations to guide a student’s early layers, and applying this guidance only in early training steps with a decaying schedule, yielding a reproducible training recipe.

* The approach is architecture-agnostic and effective with small teachers.
Because alignment is performed on hidden states, the method imposes minimal architectural constraints and remains effective even when the teacher is 10× smaller than the target model, thereby increasing practical reusability of open pretrained assets.

* It demonstrates robustness across teacher families and sizes.
Using heterogeneous small teachers (e.g., OPT-125M, Pythia-160M), LET consistently accelerates convergence and improves accuracy, indicating method-level robustness beyond a single family.

* It demonstrates practical impact under constrained compute.
Under identical token budgets, LET-1.4B surpasses a baseline 3B model in downstream performance, highlighting the advantage of better training dynamics rather than brute-force scaling.

**Weaknesses:**

* There is a dependence on teacher quality.
Although LET works with small teachers, using weak or domain-mismatched teachers may inject harmful biases into the early layers, potentially leading to negative distillation effects.
It would be better to also discuss the situations in which the proposed method does not work well.

* The breadth and strictness of baselines could be improved.
While several baselines are covered, more stringent comparisons under identical token/compute/data budgets with the latest pretraining acceleration approaches (e.g., strong online distillation or growth strategies) would further solidify the claim of superiority.

* Theoretical grounding is limited.
The paper would benefit from deeper analysis of why late-to-early (in both depth and time) works, e.g., representation geometry, optimization landscape smoothing, or gradient noise reduction, perhaps via simplified models or convergence sketches.

**Questions:**

* Reductions in convergence steps do not automatically guarantee total wall-clock or cost gains once teacher feature extraction, caching, and (in distributed setups) communication overheads are included.
Can authors provide such information?

* Evidence at 1.4B/7B is promising, but it remains unclear how late -> early alignment behaves for tens of billions to hundreds of billions of parameters, especially under architectural mismatches (e.g., LayerNorm variants, depth discrepancies).
What could the authors add regarding this point? (this does not mean to perform the experiments of large models)

* Finer-grained ablations would be valuable.
More exhaustive studies disentangling late-to-early-step vs. late-to-early-layer effects, which late teacher layer(s) to use, the student’s matched layer(s).

I am willing to update the overall scores when the authors clearly answer my concerns and questions.

---

> ### Author Response · Authors · 2025-11-21
> **Response to Reviewer 4pPU (Part I)**
>
> ## W1: Failure Mode Analysis
>
> We sincerely thank the reviewer for this valuable and practical suggestion, which will significantly enhance the comprehensiveness of our presentation. While LET demonstrates effectiveness with small models $\mathcal{T}$ in our experimental settings, we note that negative effects may occur under certain special circumstances. In the newly added Failure Mode Analysis and Layer Selection Strategies section, we present a concrete case where we train a 1B-scale model using GPT2-small and GPT2-medium as $\mathcal{T}$, which indeed produces negative effects (see below).
>
> We attribute this degradation to the potentially lower quality of GPT2's training data (with a cutoff of late 2017) compared to modern language models. Consequently, GPT2's representations fail to provide effective alignment signals. Although such scenarios are relatively rare in current practice, as GPT2 is seldom used for extraction nowadays, we believe it is important to provide this cautionary note. Our experimental results demonstrate that LET exhibits robust performance across diverse model families (e.g., SmolLM, Pythia, OPT, and LLaMA), showcasing strong cross-architecture and cross-family compatibility.
>
> **Table: Average 1-shot performance across downstream tasks when employing GPT-2 variants as the small model $\mathcal{T}$**
>
> | Method | AVG | $\Delta$ |
> |:---|:---:|:---:|
> | Baseline | 41.6 | -- |
> | RKD | 39.2 | -2.4 |
> | LET-GPT2-Small | 40.7 | -0.9 |
> | LET-GPT2-Medium | 41.1 | -0.5 |
>
> ## W2: Comparison with State-of-the-Art Acceleration Methods
>
> We sincerely thank the reviewer for this constructive suggestion. We have conducted additional experiments comparing LET with SALT [1], a recent two-stage training framework that employs knowledge distillation in the first stage followed by standard training in the second stage. Under identical data token budgets, our novel LET paradigm achieves superior performance compared to SALT. Notably, instead of exhibiting the severe instabilities observed in SALT's training curves on The Pile dataset, LET maintains a smoother optimization landscape throughout training (see Figure 4). This stability, combined with improved final performance, further validates the effectiveness of our late-to-early strategy.
>
> **Table: 1-shot performance comparison across different model sizes and methods**
>
> **Model Size = 1.4B**
>
> | Method | ARC-c | ARC-e | HS | LAMB | OBQA | PIQA | SciQ | Wino. | BoolQ | Avg. |
> |:---|:---:|:---:|:---:|:---:|:---:|:---:|:---:|:---:|:---:|:---:|
> | Baseline | 17.8 | 44.2 | **28.6** | 24.1 | 26.0 | 61.5 | 73.3 | 51.4 | 47.9 | 41.6 |
> | RKD | 18.0 | 42.9 | 27.7 | 24.8 | 26.3 | 62.4 | 63.7 | 52.3 | 54.8 | 41.4 |
> | SALT | 18.1 | 45.5 | 28.5 | 24.5 | 26.3 | 64.0 | 73.6 | 52.7 | 52.9 | 42.9 |
> | LET(67%) | 17.8 | **45.7** | 28.1 | 23.8 | 26.6 | **64.6** | 72.2 | 52.6 | 51.1 | 42.5 |
> | **LET** | **18.3** | 45.3 | 28.4 | **24.9** | **26.8** | 64.4 | **74.0** | **53.0** | **57.3** | **43.6** |
> ---
> **Model Size = 7B**
>
> | Method | ARC-c | ARC-e | HS | LAMB | OBQA | PIQA | SciQ | Wino. | BoolQ | Avg. |
> |:---|:---:|:---:|:---:|:---:|:---:|:---:|:---:|:---:|:---:|:---:|
> | Baseline | 19.4 | 45.6 | 29.3 | 25.5 | 28.0 | 63.3 | 74.5 | 52.7 | 51.4 | 43.3 |
> | RKD | 19.8 | 41.6 | 28.8 | 26.5 | 30.8 | 61.3 | 63.9 | 51.4 | 55.6 | 42.2 |
> | SALT | 19.1 | 46.8 | **30.5** | 27.4 | 30.6 | 62.1 | 76.0 | 52.9 | 56.9 | 44.7 |
> | LET(67%) | 18.4 | 45.9 | 29.5 | 27.0 | 29.7 | 61.8 | 74.1 | 51.4 | **57.3** | 43.9 |
> | **LET** | **20.0** | **47.4** | 29.8 | **28.6** | **31.4** | **65.3** | **76.7** | **54.4** | 55.9 | **45.5** |

---

> > ### Author Response · Authors · 2025-11-21
> > **Response to Reviewer 4pPU (Part II)**
> >
> > ## W3: Theoretical Insights
> >
> > We are happy to discuss on the theoretical Insights that could underpin our method from the perspective of Ordinary Differential Equations (ODEs).
> >
> > Modern LLMs, typically built upon the Transformer architecture and its variants, can be viewed as a discretized ODE where integer time steps correspond to layer indices [2]. The ODE state variables are instantiated as the intermediate hidden representations that progressively evolve across layers, transitioning from input embeddings to final output representations used for next-token prediction.
> >
> > A crucial concept in solving ODEs is that the solution space can be constrained by applying boundary or intermediate point conditions. Building on extensive research that establishes how both intermediate and final layers of LLMs encode rich, meaningful representations [3, 4, 5], we leverage this insight in our design. Our proposed LET alignment between the target model $\mathcal{M}$’s early representation and the small model $\mathcal{T}$’s final representation effectively imposes an intermediate point condition on the learning trajectory of $\mathcal{M}$. This constraint guides the optimization process, steering the representation evolution of $\mathcal{M}$ towards a semantically meaningful region of the representation space as it learns to fit the training data.
> >
> > Notably, what sets our LET paradigm apart from traditional KD is its emphasis on transferring abstract knowledge in early state, rather than simply aligning model outputs as is common in conventional KD. This distinction becomes particularly crucial in our unconventional setting of using a smaller model to train a larger one. This is because, as training progresses, the larger model $\mathcal{M}$ is expected to eventually surpass the smaller $\mathcal{T}$ in overall performance. In such settings, persisting with logit-level alignment to $\mathcal{M}$ could introduce negative effects.
> >
> > However, we know that other theoretical frameworks might also offer valid interpretations. The development of a standard and unified theoretical framework for such alignment techniques in LLMs is still an open and valuable research direction, which we leave for future work.
> >
> > [1] Rawat, Ankit Singh, et al. "A little help goes a long way: Efficient llm training by leveraging small lms." arXiv preprint arXiv:2410.18779 (2024).
> >
> > [2] Chen, Ricky TQ, et al. "Neural ordinary differential equations." Advances in neural information processing systems 31 (2018).
> >
> > [3] Dar, Guy, et al. "Analyzing transformers in embedding space." Proceedings of the 61st Annual Meeting of the Association for Computational Linguistics (Volume 1: Long Papers). 2023.
> >
> > [4] Elhoushi, Mostafa, et al. "Layerskip: Enabling early exit inference and self-speculative decoding." Proceedings of the 62nd Annual Meeting of the Association for Computational Linguistics (Volume 1: Long Papers). 2024.
> >
> > [5] Skean, Oscar, et al. "Layer by layer: Uncovering hidden representations in language models." arXiv preprint arXiv:2502.02013 (2025).

---

> ### Author Response · Authors · 2025-11-21
> **Response to Reviewer 4pPU (Part III)**
>
> ## Q1: Wall-Clock Time Speedup
>
> We appreciate the reviewer's concern about wall-clock time, which is indeed a critical practical consideration. First, under identical experimental configurations, LET achieves the performance of the 20B-token baseline using only 12.5B tokens (see Figure 1), demonstrating a substantial reduction in required training data. Regarding the overhead introduced by the additional forward pass of the small model $\mathcal{T}$, our throughput (tokens/s) is approximately $0.9848\times$ of the baseline (see Table 4).
>
> However, we emphasize that computational efficiency was a central consideration from the initial design phase of our method, which motivated our choice of small models. The compact size of $\mathcal{T}$ confers several critical advantages: faster forward propagation, minimal memory footprint, and communication overhead in distributed settings. Furthermore, it is important to note that LET incurs the $\mathcal{T}$ overhead only during the early training phase (before $S_{\text{stop}}$ step), after which the training proceeds identically to standard pretraining with zero additional computational cost. When accounting for these factors holistically, to achieve the same performance, our approach requires approximately $1.56\times$ less wall-clock time compared to the baseline, confirming substantial practical gains even after considering all computational overheads.
>
> ## Q2: Scalability and Architectural Mismatches
>
> We greatly appreciate the reviewer's insightful question regarding architectural compatibility, which is crucial given the increasing diversity of large scale open-source models in the community. We would like to clarify that architectural flexibility was a fundamental design principle of LET from the outset. Our method operates at the representation level and is inherently architecture-agnostic. Specifically, the target model $\mathcal{M}$ and small model $\mathcal{T}$ can adopt different architectural configurations, including different attention mechanisms, activation functions, hidden dimensions, intermediate dimensions, and number of layers (see Table 3 for a detailed comparison of architectural differences).
>
> ## Q3: Finer-Grained Ablation Studies
>
> Given the substantial computational cost of each training run and our limited computational budget, we have made our best effort to conduct additional ablation experiments to comprehensively explore the layer selection strategies for both $\mathcal{M}$ and $\mathcal{T}$. Building upon our previous ablation studies that validated the effectiveness of L2E (i.e., LET) over M2X alignment, we now investigate more granular layer-wise pairing strategies. As illustrated in Figure 12, our empirical results reveal that aligning the last layer representation of $\mathcal{T}$ with the third layer of $\mathcal{M}$ yields optimal performance. One possible explanation is that the first and second layers of $\mathcal{M}$ primarily encode input-specific information, whereas deeper layers (starting from the third layer) begin to capture more abstract knowledge representations. Overall, these results consistently demonstrate that the L2E design principle achieves superior effectiveness compared to alternative alignment strategies.

---

> > ### Comment · Reviewer_4pPU · 2025-11-27
> >
> > Thank you for providing the detailed response to my concerns and questions in my initial review.
> >
> > W1/W2:  Thank you for providing the results of additional experiments on the failure case and SOTA acceleration methods. I believe adding these results strengthen the paper.
> >
> > W3: I respectfully disagree that the authors' rebuttal successfully provides theoretical insights into the proposed method.
> > The discussion provided in the rebuttal may be correct, but it is so abstract that it does not offer any clear relationships or correlations.
> > The authors need to tighten their explanations and the method's properties using mathematical notation.
> > Moreover, the authors do not provide theoretical insights into why the final layer in smaller models improves performance when the method adds an auxiliary loss term based on cosine similarity with an early layer of the target model, rather than with the middle or final layers.
> > At lease, such theoretical insights should be discussed in the main text in the paper.
> >
> >
> > Q1: Regarding the wall-clock time, I understand the authors' rebuttal. However, one additional request is that it would be better to clearly explain what the authors wrote in this rebuttal into the main text (not in the appendix) of the paper if the space is allowed.
> >
> > Q2: Thank you for providing a clear answer.
> > However, I am not fully convinced by the authors' rebuttal.
> > In particular, I would like to focus on the potential adverse effect associated with the total number of layers in the target model.
> > Intuitively, the effect of the proposed method, LET, may decrease as the total number of layers increases, since the dominance of a single layer diminishes in larger models.
> > In this paper, the models appear to have only 24 or 32 layers, but there are also models with 80 and 126 layers, such as Llama 3.3 70B and Llama 3 405B, respectively, which are representative of larger-scale models.
> > It is necessary to provide evidence that increasing the total number of layers in models does not reduce the effectiveness of the proposed method.
> > For example, the authors could infer the absence of an adverse effect based on the empirical results from smaller models.
> >
> >
> > Q3: I fully understand the limited computational budget. However, the authors could conduct more comprehensive ablation studies on much smaller models, such as several-million-parameter models. Such a setting would not significantly affect the computational budget.
> > Ideally, it should be addressed, but I also understand that there is not enough time to do so before the rebuttal period ends. Therefore, I will not consider this point to be a critical flaw of this paper at this stage of the review.
> >
> >
> > Overall, my concerns remain. I will maintain my score for now. I remain open to revisiting my overall assessment if the authors provide a further rebuttal.
> >
> > ---
> > If I understand correctly, according to the ICLR 2026 Author Guide, authors are allowed to add additional pages during the rebuttal period:
> >
> > > During the discussion/rebuttal phase and for the camera ready, the page limit will be increased to 10 pages to allow for new results/discussions.
> >
> > This is not mandatory, but based on this guidance, rebuttals that are crucial to this paper's discussion should be incorporated into the main body rather than placed in the appendices.
> > In particular, it would be better to add a new subsection in the discussion section and provide theoretical insights of the proposed method.
> > Moreover, potentially, the additional experimental results should be added to the main text.

---

> ### Author Response · Authors · 2025-11-27
>
> Dear Reviewer 4pPU,
>
> I hope this message finds you well.
>
> As the discussion phase nears its end, we would like to ensure that our responses have sufficiently addressed all your concerns. We remain available to answer any further questions or discuss additional feedback you may have.
>
> We truly value your insights and have incorporated your suggestions to improve our manuscript.
>
> Thank you again for your time and constructive feedback.

---

> ### Author Response · Authors · 2025-12-03
>
> We sincerely appreciate your constructive comments, which have enhanced the comprehensiveness of our presentation.
>
> W1/W2: We are glad to learn that our responses and additional experiments regarding the failure mode analysis and recent acceleration methods have effectively addressed your concerns.
>
> W3: We sincerely appreciate your constructive suggestions regarding the theoretical framing of our method. Following your advice to utilize simplified models for discussion, we have conducted a theoretical analysis focusing on the optimization landscape. We formally prove that LET achieves a smaller Frobenius norm upper bound, which implicitly constrains the spectral norm and the associated Lipschitz smoothness constant.
>
> For the complete proof, please refer to the Theoretical Analysis section in the updated manuscript. Our analysis demonstrates that, compared to a larger $k$, LET incurs a smaller theoretical upper bound on the additional curvature cost, thereby preserving more of the original optimization landscape.
>
> While this analysis on simplified models solidifies our theoretical grounding, fully characterizing the optimization dynamics of modern, complex architectures remains a valuable open problem, which we plan to explore in future work.
>
> Q1: We appreciate your valuable feedback. We fully recognize the importance of integrating these clarifications into the main paper. Taking advantage of the allowed page limit extension, we commit to moving the most important discussions, including the wall clock time analysis and the new theoretical insights, from the appendix to the main text in the final version of the paper.
>
> Q2/Q3: We are grateful for your understanding of our computational constraints and your insightful suggestion to infer the scalability of our method through empirical trends in smaller models.
>
> Our previous experiments primarily focused on the 1B and 7B scales (corresponding to 24 and 32 layers, respectively). To address your concern regarding the potential diminishing effect of our method in deeper models, we conducted an additional ablation study on a 160M model with only 12 layers. This allows us to observe the trend of effectiveness across varying depths. The results are summarized below:
>
> | $\mathcal{M}$ Size | Num Layers | $\mathcal{T}$ Size | Performance Gain |
> |:------------------------:|:----------:|:------------------------:|:---------------------------:|
> | 160M                     | 12         | 135M                     | +2.6                   |
> | 1.4B                     | 24         | 135M                     | +2.0                    |
> | 7B                       | 32         | 1.7B                     | +2.2                    |
>
> As shown above, LET demonstrates consistent performance gains across models with 12, 24, and 32 layers. This consistency suggests that the effectiveness of LET is robust to variations in model depth, supporting the inference that the method remains effective for deeper architectures.
>
> Thank you again for your constructive comments. Your feedback has significantly strengthened the comprehensiveness of our presentation. We are encouraged by your acknowledgement that our new results have strengthened the paper, and we hope these additional theoretical insights and empirical validations further solidify your assessment of our work.

---

### Author Response · Authors · 2025-11-21
**Appreciation to all Reviewers**

### Dear Reviewers,

We would like to extend our sincerest gratitude to all of you for your time and dedicated effort in reviewing our manuscript. We deeply appreciate your insightful comments and constructive suggestions.

We are encouraged to see that the reviewers recognize the **comprehensive and well-designed experiments** (Reviewers 4pPU, dPjE, srWB), the **novelty of LET** (Reviewer srWB), the **practical significance** of our method LET, including its architecture-agnostic design, effectiveness even with small model alignment, and robustness across diverse families (Reviewers 4pPU, ec8P, dPjE), and the **clarity of our presentation** (Reviewers 4pPU, ec8P, dPjE, srWB).

In response to your valuable feedback, we have made the following revisions to our manuscript:

- Added comparison experiments with SALT in Table 1
- Conducted failure mode analysis
- Performed fine-grained ablation studies on layer selection
- Provided detailed analysis of throughput, wall-clock time, and peak VRAM
- Conducted in-depth investigation of alignment methods, including CCA, logsum, and tokenizer-mismatch settings
- Expanded the discussion of prior works

We have given careful consideration to all the comments raised and have responded to each of them in the clear and detailed response below. To facilitate your review of the revised manuscript, we have highlighted all major revisions and newly added experimental results in blue.

We believe these revisions have substantially strengthened the paper and effectively addressed your concerns. We sincerely look forward to your feedback and welcome any further discussion.


Best regards,

Submission 5787 Authors

---

### Meta-Review · Area_Chair_sMDP · 2025-12-23

**Summary:**

1. The performance depends on the teacher's quality. Using weak or domain-mismatched teachers may inject harmful biases into the early layers, potentially leading to negative distillation effects. (Reviewer 4pPU)
2. This paper omits some SOTA baselines. (Reviewer 4pPU)
3. This paper lacks the necessary theoretical grounding. (Reviewer 4pPU, dPjE, and srWB)
4. This paper misses further analysis with respect to prior works in KD literature, and the novelty of the problem statement is a bit overclaimed. (Reviewer srWB)
5. The concerns about the generalization of models with hundreds of billions of parameters. (Reviewer 4pPU)
6. Lacking finer-grained ablations (e.g., disentangling late-to-early-step vs. late-to-early-layer effects, which late teacher layer(s) to use, the student’s matched layer(s).) (Reviewer 4pPU)

**Reviewer Concerns:**

The authors have addressed almost all the reviewers' questions during the rebuttal phase. It is recommended to incorporate these experimental results into the new version.

However, the theoretical analysis seems too simple, and the organization of the paper may need to be further refined. (Reviewer 4pPU)

**Reviewer Scores:**

Reviewer 4pPU: retains 4

~~Reviewer ec8P: 4~~  (I have ignored this reviewer's comments since they contained some factual errors.)

Reviewer dPjE: retains 8

Reviewer srWB: retains 6

---

### Decision · Program_Chairs · 2026-01-26

Accept (Poster)